# Distillation Lineage Inspector: Black-Box Auditing of Model Distillation in LLMs

## Abstract

Model distillation has emerged as a widely used technique for creating efficient models tailored to specific tasks or domains. However, its reliance on knowledge from foundation models raises significant legal concerns regarding intellectual property rights. To address this issue, we propose the Distillation Lineage Inspector (DLI) framework, which enables model developers to determine whether their large language models (LLMs) have been distilled without authorization, even in black-box settings where training data and model architecture are inaccessible. DLI is effective across both open-source and closed-source LLMs. Experiments show that DLI achieves 80% accuracy with as few as 10 prompts in fully black-box settings and yields a 45% improvement in accuracy over the best baseline under standard experimental conditions. Furthermore, we analyze how auditor knowledge of target models influences performance, providing practical insight for building privacy-preserving and regulation-compliant AI systems. Our code is open-sourced at `https://anonymous.4open.science/r/DLI-0DCC`.

## 1 Introduction

Model distillation is a central technique for scaling LLM deployment, mitigating large model size, high compute cost, and slow inference Gou et al. (2021). By transferring knowledge from a large and highly effective teacher model to a smaller and more efficient student model, distillation enables the preservation of accuracy while allowing for use in resource-constrained settings Xu et al. (2024). Recent advances in distillation techniques have further enhanced the effectiveness of this process. For example, the Chinese startup DeepSeek distills large models, such as Qwen2.5-32B Team (2024), to produce students, including DeepSeek-R1-Distill-Qwen-32B, which match or surpass OpenAI's o1-mini on tasks like mathematical reasoning, code generation, and commonsense reasoning Team (2024); Guo et al. (2025). Likewise, OpenAI has distilled smaller models such as (e.g., GPT-4o-mini) using outputs from larger teacher models, such as GPT-4o, to achieve task-comparable quality at a lower cost OpenAI (2024). These innovations illustrate how distillation compresses state-of-the-art language models into lightweight variants suitable for real-world deployment.

However, reliance on teacher-derived knowledge raises legal and intellectual property concerns. For instance, an AP News report Maekawa et al. (2025) relays statements from OpenAI and a senior White House AI adviser suggesting that DeepSeek may have used ChatGPT outputs to train its models, a process analogous to distillation and potentially at odds with OpenAI terms of service and intellectual property policies OpenAI (2025a;b) This example illustrates that unauthorized knowledge transfer can create material contractual and intellectual property liabilities. As LLMs become increasingly prevalent, consider a scenario in which a startup distills a specific functionality (e.g., summarization) from a large proprietary model and deploys it as a paid service; such use may violate the terms of the provider. Provenance auditing is therefore essential for protecting intellectual property and supporting the sustainable development of large-scale AI models.

In this paper, we focus on auditing a black-box model to determine whether it was distilled from a specific teacher model. Figure 1 depicts an audit scenario with multiple suspicious models. Auditing these models poses three key challenges. First, an auditor's access is typically limited to *black-box* settings, often to probabilities or only generated tokens, which challenges provenance inference Nikolic et al. (2025). Second, distilled provenance is challenging because teacher and student architectures may differ significantly, unlike provenance tracing for pre-trained and fine-tuned

Figure 1: A system model for auditing suspicious models.

models Nikolic et al. (2025); Foley et al. (2023), which prevents straightforward feature comparisons. Third, differences in domain or task adaptation further complicate provenance detection; for example, student models distilled for specialized domains (e.g., healthcare, legal, finance) can shift output distributions away from teacher behavior Xu et al. (2024). Existing work for distilled provenance Wadhwa et al. (2025) relies on syntactic Part-of-Speech (PoS) templates Shaib et al. (2024), which are often unreliable and achieve low accuracy. Moreover, the problem is typically cast as a *closed-world* identification task, in which the auditor must choose a teacher from a set of fixed candidates Wadhwa et al. (2025). In practice, however, model developers and auditors typically require *open-world* verification, making the closed-world assumption unrealistic.

To move beyond closed-world assumptions, we propose the **D**istillation **L**ineage **I**nspector (**DLI**) framework for distilled model auditing in black-box settings. For realism, we further consider two observability settings: **Token-Level Access** ($\mathcal{K}_t$), where the auditor observes generated tokens and their probabilities, and **Word-Level Access** ($\mathcal{K}_w$), where only the final text is visible. DLI requires no knowledge of the teacher/student architectures or training data. It achieves this by constructing shadow-distilled proxies to mimic suspicious behavior. These shadow models enable the generation of a labeled dataset, which is then used to train an auditing model. DLI identifies model-style tokens to quantify the extent to which teacher models embed distinctive "signatures" into student models during distillation. This design effectively distinguishes students of a hypothesized teacher from other suspects even under black-box constraints and directly answers the practical question: **Is the suspicious model distilled from the target LLM?**

To evaluate the efficacy of the proposed DLI framework, we conducted extensive experiments across diverse architectures, including open-source models such as Llama-3.1-8B and closed-source models such as GPT-4o-mini, as teachers, and a pool of ten domain-specific distillations with their original counterparts as student and non-student suspects. We audit models suspected of being distilled from Llama-3.1-8b using HealthCareMagic-100k dataset. Under $\mathcal{K}_t$, DLI achieved 100% accuracy and 100% in F1, markedly exceeding baselines. Under the more restrictive $\mathcal{K}_w$, DLI still reaches the 85% accuracy and 86% in the F1. We further evaluate on the QA Legal dataset to assess generalizability. The consistent trends across these diverse settings demonstrate the adaptability and robustness of the DLI framework across architectures, observability settings, and domains.

Our main contributions are as follows:

- We initiate the study of auditing the provenance of *distilled* models, framing open-world lineage verification as a principled auditing task.
- We introduce the *Distillation Lineage Inspector* (DLI), a model-agnostic auditing framework for accurate provenance verification under two access regimes: *Token-Level Access* (with log-probabilities) and *Word-Level Access* (text-only).
- We develop a model-style token selection algorithm within DLI, enabling effective distillation detection even under limited or black-box access.
- We evaluate DLI on both open-source and closed-source teacher models, including heterogeneous and domain-specific distillation. Compared to the SOTA baseline, DLI improves accuracy by 45% under $\mathcal{K}_t$ and 40% under $\mathcal{K}_w$, showing effectiveness across observability settings.

## 2 RELATED WORK

**Data Provenance Tracing** Data provenance lies within the broader provenance literature, where the core problem involves auditing whether a specific piece of data was used to train a machine

learning model Hisamoto et al. (2020); Song & Shmatikov (2019); Zeng et al. (2024b) or a retrieval-augmented generation (RAG) system Zeng et al. (2025). The task is typically framed as a membership inference attack (MIA). MIAs pose significant privacy risks and often underlie stronger threats, such as data extraction attacks Carlini et al. (2021); Panchendrarajan & Bhoi (2021); Zeng et al. (2024a); Huang et al. (2022); Zeng et al. (2024c). Given their strong connection to privacy concerns, MIAs have become a widely adopted tool to quantify vulnerabilities in both classical models Shokri et al. (2017); Jagielski et al. (2023); Yeom et al. (2018) and large language models (LLMs) Mireshghallah et al. (2022); Mattern et al. (2023); Debenedetti et al. (2023). However, these methods primarily address data-level provenance, whereas our focus is model-level provenance.

**Model Provenance Tracing** Methods for establishing model ownership and detecting illegitimate use are increasingly important. A prevalent strategy embeds watermarks in the model Jia et al. (2021); Shao et al. (2024); Yan et al. (2023). A watermarked model produces a specific output in response to a carefully crafted input, enabling attribution to its original creator. This mechanism provides a means of protecting proprietary models through designed input–output pairs that act as ownership signatures. However, watermarking requires control of the training process, which can degrade utility. We provide additional discussion in Appendix A.6.

Meanwhile, the rise of knowledge transformation techniques, such as fine-tuning and distillation, has intensified concerns about copyright and ownership. Recent studies have proposed various methods for model provenance tracing in natural language processing Wadhwa et al. (2025); Foley et al. (2023); Nikolic et al. (2025). For example, Foley et al. (2023) studies LLM attribution by linking a fine-tuned model to its pre-trained base via embeddings of generated tokens. Similarly, Nikolic et al. (2025) develops a framework for testing whether one model is derived from another, grounded in the observation that real-world derivations preserve detectable similarities in model outputs.

Existing methods on provenance-tracing primarily focus on mapping a fine-tuned LLM back to its pre-trained base, not distillation, where student architectures and capacities may diverge substantially from the teacher. As a result, distillation-focused provenance has received relatively little attention in prior research. In addition, many approaches also assume strong priors or auxiliary data (e.g., prompts from the test model's training set Foley et al. (2023)), which limits practicality. Most relevant to our work, Wadhwa et al. (2025) leverages syntactic Part-of-Speech (PoS) templates Shaib et al. (2024) to identify teacher models via higher-order lexical features under distillation. However, these features neither capture deeper teacher–student similarity (as their results indicate) nor escape the closed-world verification assumption. We relax this assumption by formulating provenance as an open-world verification problem, yielding a more realistic and broadly applicable setting.

## 3 PRELIMINARY

### 3.1 DISTILLATION WITH LLMS

**Model-level Distillation** Knowledge distillation transfers knowledge from a large teacher model $T$ to a smaller student model $S$ Jiao et al. (2020); Sanh et al. (2019). Given an input $x$, the teacher defines a token distribution at position $t$ as $p_T(\cdot \mid x, t) = \mathrm{softmax}(z_T(x, t))$, where $z_T(x, t)$ are the teacher logits. The student analogously defines $p_S(\cdot \mid x, t) = \mathrm{softmax}(z_S(x, t))$. Distillation minimizes a weighted combination of the cross-entropy loss between teacher and student token distributions and the standard supervised cross-entropy loss:

$$\mathcal{L} = \alpha\Big(-\sum_{t=1}^{T}\sum_{v=1}^{V} p_T(v \mid x, t)\log p_S(v \mid x, t)\Big) + (1-\alpha)\Big(-\sum_{t=1}^{T}\log p_S(y_t \mid x, t)\Big), \quad (1)$$

where $\alpha \in [0, 1]$ balances the distillation and supervised objectives. This encourages the student to match teacher behavior beyond hard labels, capturing richer structural information.

**Data-level Distillation** Data-level distillation aims to train a smaller distilled model to replicate the performance of a larger teacher model Maekawa et al. (2025); Liu et al. (2022); Gallagher (2023). Rather than mimicking teacher outputs, this approach focuses on creating a distilled corpus from teacher predictions and training the student on it. Formally, given a large dataset $\mathcal{D} = \{(x_i, y_i)\}_{i=1}^{N}$, the teacher model $T$ makes predictions for each input $x_i$ and produces soft labels $\hat{y}_i = T(x_i)$, resulting in a distilled corpus $\mathcal{D}_{\mathrm{distill}} = \{(x_i, \hat{y}_i)\}_{i=1}^{N}$, where the labels are derived from the teacher

rather than from the ground-truth. The distilled model $S$ is then optimized to match the teacher's predictions $\hat{y}_i$, with a standard loss function $\ell$, typically cross-entropy:

$$\mathcal{L}_{\text{distill}} = \sum_{i=1}^{N} \ell\big(S(x_i), \hat{y}_i\big). \tag{2}$$

This performance matching paradigm transfers teacher knowledge to the student while preserving task performance. Training on a teacher-generated distilled corpus yields compact, efficient models suitable for deployment in resource-constrained settings such as mobile devices or edge computing.

## 3.2 PROBLEM FORMULATION

We study provenance auditing for distilled models in realistic scenarios where a third-party seeks to verify whether a deployed model was derived by distillation from a proprietary LLM. Figure 1 illustrates the overall auditing framework, which involves three core entities:

- **Model Developer:** owns and trains a proprietary teacher model $T$ which may be public or private, while retaining intellectual property (IP) rights over the model.
- **Suspected Deployer:** serves a model $M$ that provides API access to its outputs. The model $M$ may be a student model secretly distilled from $T$, raising concerns about unauthorized reuse.
- **Auditor:** an external party that interacts with $M$ via its API to test if $M$ was derived from $T$.

This setting reflects practical auditing needs, from verifying unauthorized reuse of proprietary models by third-party providers to detecting potential IP violations in distillation pipelines.

**Audit Objective.** Given black-box access to a target teacher model $T$ and a suspicious model $M$, the auditor aims to determine whether $M$ was obtained by distilling from $T$, i.e., to decide whether:

$$\texttt{IsDistilled}(M,T) = \begin{cases} 1, & \text{if } M \text{ is distilled from } T, \\ 0, & \text{otherwise.} \end{cases}$$

**Auditor's Capabilities.** We assume a strict black-box setting: no access to architectures, parameters, training procedures, or distillation data for $T$ and the suspicious model $M$. Interaction is limited to an API that returns predictions $P_M(x)$ for queries $x$. The auditor holds a query set $\mathcal{X} = \{x_1, \ldots, x_n\}$ to probe both the suspicious model $M$ and the teacher model $T$. The auditor can train shadow models distilled from $T$, and build an auditor model $M_{audit}$ that predicts whether $M$ originates from $T$. This constraint mirrors real deployments and emphasizes robustness. Based on these assumptions, we further consider two observability settings available to the auditor.

- **Token-Level Access** ($\mathcal{K}_t$)**.** Under this setting, the auditor can observe both the generated tokens along with their output probabilities.
- **Word-Level Access** ($\mathcal{K}_w$)**.** In this more restrictive setting, the auditor has access only to the final generated text without any probability information.

This setup reflects realistic provenance auditing conditions and supports broader applications, such as intellectual property protection, accountability in model deployment, and compliance verification.

## 4 AUDITING MODEL LINEAGE

Our methodology is grounded in a key observation: *Despite architectural or implementation divergence, it often preserves behavioral similarities because the distillation objective aligns its outputs with the teacher.* When probed with the same input, the student reproduces not only similar predictions but also stylistic patterns characteristic of the teacher. These consistencies focus on specific high-probability lexical or syntactic tokens, which we refer to as *model-style tokens*. By identifying and quantifying such tokens across diverse queries, we infer whether a target model was distilled from a given teacher, even in a strict black-box setting without access to the model's internals.

### 4.1 OVERVIEW

Building on this insight, we introduce the **D**istillation **L**ineage **I**nspector (**DLI**) framework for model provenance auditing, as illustrated in Figure 2. DLI comprises five stages: (1) **Building Distilled Proxies**, in which shadow student models are distilled from the teacher to mimic plausible deriva-

Figure 2: Architecture of the proposed Distillation Lineage Inspector (DLI).

tions; (2) **Identifying Model-Style Tokens**, which captures distinctive teacher output patterns; (3) **Encoding Behavioral Signatures**, where the shadow student and shadow non-student models are probed with shared inputs and encode their responses on model-style tokens into distinguishing behavioral signatures as feature vectors; (4) **Learning Distillation Signatures**, where a binary classifier is trained on the extracted features to separate teacher derived models from others; and (5) **Inferring Model Lineage**, where the trained auditor is applied to assess whether a suspicious model was likely distilled from the teacher based on behavioral similarity. Crucially, the DLI framework does not require access to the internal model architectures, parameters, or training data, and is thus applicable to a wide range of black-box LLMs, regardless of diverse sizes, structures, and domains.

## 4.2 BUILDING DISTILLED PROXIES

The auditor first constructs negative examples by selecting models $M_n$ that are not distilled from the target teacher model $T$. Shadow student models $M_s$ are then distilled from $T$ to serve as positive examples, representing models plausibly derived from the teacher. This setup enables the auditor to learn to distinguish between student models and unrelated models while controlling for architectural similarity. To prevent confounding from architectural choices, we employ an ensemble of multiple shadow proxies across diverse architectures and capacities (Section 5). This architectural diversity is crucial for real-world auditing, where the auditor is unaware of the suspicious model's architecture.

## 4.3 IDENTIFYING MODEL-STYLE TOKENS

Beyond matching predictions, a distilled student shows recurring lexical and syntactic preferences to which the teacher assigns high probability. We call these high-probability continuations *model-style tokens* and use them as sparse discriminative features for lineage tracing. For each input $x \in \mathcal{X}$, the auditor splits the sequence into equal-length prefix and suffix. The prefix is used to query the teacher $T$, yielding continuations $s$ that share the prefix of $x$ but differ in the suffix. This produces a new generated dataset $D_g$. To separate teacher-specific preferences from general language regularities, we introduce a generic comparison model $M_c$. The suffix of $s$ is disclosed incrementally, with $M_c$'s next-token probabilities recorded at each step. To identify model-style tokens, we rank suffix tokens based on their prediction difficulty, defined as the inverse of their predicted probabilities. Intuitively, suffix tokens that the target readily produces but $M_c$ assigns low probability are considered difficult and flagged as candidate model-style tokens. Let the selection factor $k$ indicate the retained proportion. Finally, we rank suffix tokens by difficulty and select the top $1/k$ portion as model style tokens, which are aggregated into $D_{\text{tokens}}$. Algorithm 1 summarizes the procedure. This comparator-based ranking normalizes target-specific likelihood scales and emphasizes teacher-specific stylistic biases.

## 4.4 ENCODING BEHAVIORAL SIGNATURES

After constructing distilled proxies, the auditor extracts behavioral features that capture how different models respond to the same input. This is done by probing the shadow student models and the shadow non-student models using the model-style tokens dataset $D_{\text{tokens}}$.

**Behavioral Probing Setup** Before querying the models with model-style tokens, we form prefix and target pairs from each token list $U_i \in D_{\text{tokens}}$ corresponding to an original input sample $s_i =$

$(w_1, w_2, \ldots, w_n)$. For each token $t_j \in U_i$ with position $p_j$ in $s_i$, we define the probing prefix as $x_j = (w_1, w_2, \ldots, w_{p_j-1})$. We query the model with $x_j$ to get the next token distribution $P(\cdot \mid x_j)$ and record the probability of the target token $t_j$ as :

$$P_j = P(t_j \mid x_j). \tag{3}$$

This process is repeated for all $t_j \in U_i$ and all $U_i \in D_{\text{tokens}}$, yielding context-aligned responses that quantify each model's propensity to produce model-style tokens. Under $\mathcal{K}_t$, the auditor observes generated tokens and their probabilities, which are available for many open source models and some commercial APIs, including the GPT series. To stress test robustness, we consider a more restricted setting $\mathcal{K}_w$, where the auditor observes only generated words, without access to token-level probabilities. In this case, next token probabilities are estimated by sampling. Specifically, for each probing prefix $x_j$, we query the model $N$ times (with $N = 5$ in our experiments) and let $c_j$ be the number of trials in which the next token equals the target $t_j$. The empirical probability is then:

$$\hat{P}_j(t_j \mid x_j) = \frac{c_j}{N}. \tag{4}$$

This frequency-based estimate $\hat{P}_j(t_j \mid x_j)$ serves as a proxy for the true probability of the next token when token-level signals are unavailable.

**Feature Construction**   Next, we construct feature vectors that capture the behavioral responses of the models to model-style tokens. For each sample $s$ with $U \in D_{\text{tokens}}$, let $P^{(M)}(s) = \{p_1^{(M)}, p_2^{(M)}, \ldots, p_k^{(M)}\}$ denote the set of predicted probabilities returned by model $M \in \{M_s, M_n\}$ for the next token predictions on each model-style token associated with sample $s$, where $p_i^{(M)} \in [0, 1]$ is the next token probability that model $M$ assigns to the target token $t_i$, and $k = |U|$ is the number of model-style tokens extracted from $s$. To standardize these predictions into fixed-size feature vectors, we partition the probability range $[0, 1]$ into $m$ intervals:

$$I_j = \left[ \frac{j-1}{m}, \frac{j}{m} \right), \quad j = 1, 2, \ldots, m. \tag{5}$$

Each probability $p_i^{(M)}$ is then assigned to an interval via $j = \left\lfloor m \cdot p_i^{(M)} \right\rfloor + 1$. We define the feature vector $F^{(M)}(s) \in \mathbb{R}^m$ as a histogram by counts over these intervals:

$$F^{(M)}(s) = (f_1^{(M)}, f_2^{(M)}, \ldots, f_m^{(M)}), \tag{6}$$

where each component counts the number of predictions that fall into the interval $I_j$:

---

**Algorithm 1:** Model-style Tokens Selection

**Input:** generated dataset $D_g$, selection factor $k$

**Output:** $D_{\text{tokens}}$

$D_{\text{tokens}} \leftarrow \emptyset$;

**for** $s_i \in D_g$ **do**
    $P_{\text{token}} \leftarrow \emptyset$;
    $U \leftarrow \emptyset$;
    prefix $\leftarrow s_i[\text{prefix}]$;
    suffix $\leftarrow s_i[\text{suffix}]$;
    $\ell \leftarrow \text{len}(\text{suffix})$;
    c_prefix $\leftarrow$ prefix;
    **for** $i \leftarrow 1$ **to** $\ell$ **do**
        $p_i \leftarrow M_c(\text{suffix}[i] \mid \text{c\_prefix})$;
        **append**$(P_{\text{token}}, p_i)$;
        c_prefix $\leftarrow$ c_prefix $\parallel$ suffix$[i]$;
    $n \leftarrow \lfloor \frac{\ell}{k} \rfloor$;
    $I_{min} \leftarrow$ **argsort**$(P_{\text{token}})[: n]$;
    **for** $i \in I_{min}$ **do**
        **append**$(U, \text{suffix}[i])$;
    **append**$(D_{\text{tokens}}, U)$

**return** $D_{\text{tokens}}$;

---

$$f_j^{(M)} = \left| \left\{ p_i^{(M)} \in P^{(M)}(s) \mid p_i^{(M)} \in I_j \right\} \right|. \tag{7}$$

For each sample $s$, we thus extract two feature vectors: $F^{(M_s)}(s)$ labeled as "member", and $F^{(M_n)}(s)$ labeled as "non-member". This yields the final audit dataset:

$$D_{\text{audit}} = \left\{ \left( F^{(M_s)}(s), 1 \right), \left( F^{(M_n)}(s), 0 \right) \mid s \in D_g \right\}. \tag{8}$$

### 4.5 Learning Distillation Signatures

We train a binary provenance classifier $M_{\text{audit}}$ on the audit dataset $D_{\text{audit}}$ using feature vectors derived from both shadow student and non-student models. These features capture fine-grained behavioral signals indicative of distillation lineage. The task is supervised binary classification: given $F^{(M)}(s)$ for a suspicious model $M$, the classifier learns to determine whether model $M$ belongs to the teacher lineage, with label 1 for teacher-derived students and 0 for non-students of teacher $T$. To achieve this, we leverage AutoGluon Erickson et al. (2020), a state-of-the-art AutoML framework that au-

tomates model selection and hyperparameter optimization. The final model $M_{\text{audit}}$ is selected by performance on a held-out validation set, yielding a robust and accurate auditor.

### 4.6 INFERRING MODEL LINEAGE

The auditor tests whether the suspicious model was distilled from the target by querying it with probing prefixes reconstructed from the model-style token set $D_{tokens}$. For each test sample $s_i$, the auditor collects the next token probabilities $P^{(M)}(s_i)$, and encodes them as $v_i = F^{(M)}(s_i)$ using the same mapping as in training. The feature vector $v_i$ is then fed to the trained auditor $M_{\text{audit}}$, which returns a binary decision indicating consistency with the target distillation signature. Overall provenance is decided by majority voting over sample-level predictions, a standard ensemble rule Kuncheva (2014). If a majority are pos-

| ID | Base Model | Params |
|----|------------|--------|
| 0 | GPT-Neo-1.3B | 1.37B |
| 1 | Phi-4-mini-Instruct | 3.84B |
| 2 | OPT-1.3B | 1.30B |
| 3 | Qwen2-0.5B | 0.49B |
| 4 | GPT2-Large | 0.81B |
| 5 | EXAONE-3.5-2.4B-Instruct | 2.41B |
| 6 | Instella-3B | 3.11B |
| 7 | Starcoder2-3B | 3.03B |
| 8 | SmolLM2-1.7B | 1.71B |
| 9 | Aprel-5B-Base | 4.83B |

Table 1: Summary of the distilled models, their base models, and parameter counts.

itive, we conclude the suspicious model was very likely distilled from the teacher. This inference process enables practical provenance auditing in black-box scenarios without parameter or data access, applicable under both observability settings for open-source and closed-source APIs.

## 5 EXPERIMENTS

### 5.1 EXPERIMENT SETUP

**Datasets.** We evaluate on two public domain-specific benchmarks. HealthCareMagic 100k[1] contains 112,165 patient–doctor conversations from HealthCareMagic. QA-Legal[2] contains legal question–answer pairs from online forums covering housing, contracts, employment, and immigration.

**Models.** To diversify the evaluation pool, we utilize 10 pre-trained LLMs to generate corresponding distilled models, as shown in Table 1. These models vary in architecture, parameter size, and training objectives, allowing us to capture a broad spectrum of model behaviors. For the target teacher models, we select four widely adopted large language models: `Llama-3.1-8B` (LLaMA3), `pythia-12b-v0` (Pythia), `Qwen2.5-7B` (Qwen2.5) and `GPT-4o-mini` (GPT-4o).

**Baselines.** We compare our method against the following model provenance tracing strategies, each designed to address knowledge transfer techniques such as fine-tuning and distillation:

- **Matching Pairs (MP)** Foley et al. (2023): This method leverages embeddings of prompts and responses to capture the characteristics of the base model. A classifier is then trained on these representations to perform model attribution.
- **Model Provenance Testing (MPT)** Nikolic et al. (2025): This approach employs multiple hypothesis testing to assess model similarity, comparing the control model against a baseline distribution derived from unrelated models.
- **Part-of-Speech Templates (PoS)** Wadhwa et al. (2025): This technique uses syntactic part-of-speech templates as higher-order linguistic features that can capture subtle stylistic signals originating from the teacher model and retained in the distilled student model.

**Evaluation Metrics.** For each suspicious model, we infer its lineage and verify whether the prediction is correct. We report accuracy and F1 score, which are standard metrics in classification tasks. These metrics are computed based on the number of true positives (TP) and true negatives (TN), representing correctly identified distilled and non-distilled models, respectively.

**Implementation Details.** Our experiments are conducted on an NVIDIA A100 GPU. For model-style token selection, we use GPT 2 XL Radford et al. (2019) with 1.61B parameters as the compar-

---

[1] https://huggingface.co/datasets/RafaelMPereira/HealthCareMagic-100k-Chat-Format-en

[2] https://huggingface.co/datasets/ibunescu/qa_legal_dataset_train

| Dataset | Method | LLaMA3 | | GPT-4o | | Pythia | | Qwen2.5 | |
|---|---|---|---|---|---|---|---|---|---|
| | | Accuracy | F1-score | Accuracy | F1-score | Accuracy | F1-score | Accuracy | F1-score |
| HealthCare | MP | 0.55 | 0.47 | 0.70 | 0.57 | 0.65 | 0.67 | 0.70 | 0.75 |
| | PoS | 0.55 | 0.69 | 0.60 | 0.67 | 0.50 | 0.00 | 0.35 | 0.52 |
| | MPT | 0.40 | 0.57 | 0.65 | 0.46 | 0.15 | 0.00 | 0.55 | 0.18 |
| | Ours($\mathcal{K}_t$) | **1.00**↑45% | **1.00**↑31% | **0.90**↑20% | **0.89**↑22% | **0.95**↑30% | **0.95**↑28% | **0.85**↑15% | **0.82**↑7% |
| | Ours($\mathcal{K}_w$) | 0.85↑30% | 0.86↑17% | **0.90**↑20% | **0.89**↑22% | 0.90↑25% | 0.89↑22% | 0.75↑5% | 0.67 |
| QA-Legal | MP | 0.55 | 0.61 | 0.85 | 0.82 | 0.45 | 0.20 | 0.55 | 0.18 |
| | PoS | 0.50 | 0.67 | 0.70 | 0.75 | 0.20 | 0.20 | 0.55 | 0.67 |
| | MPT | 0.30 | 0.46 | 0.75 | 0.74 | 0.20 | 0.00 | 0.50 | 0.00 |
| | Ours($\mathcal{K}_t$) | 0.80↑25% | 0.75↑8% | **1.00**↑15% | **1.00**↑18% | **1.00**↑55% | **1.00**↑80% | **0.95**↑40% | **0.95**↑28% |
| | Ours($\mathcal{K}_w$) | **0.95**↑40% | **0.95**↑28% | **1.00**↑15% | **1.00**↑18% | 0.90↑45% | 0.89↑69% | 0.70↑15% | 0.57 |

Table 2: Overall evaluation of auditing methods. Best results are **bold**; second best are underlined.

ison language model. The primary task is text generation, where the model is required to complete or continue a given input sequence—a fundamental capability of large language models. Following Wadhwa et al. (2025), training data are generated by the teacher models. To distill compact student models in a parameter-efficient manner, we adopt Low-Rank Adaptation (LoRA) Hu et al. (2022).

## 5.2 OVERALL EVALUATION

Table 2 summarizes the results for Accuracy and F1 scores, demonstrating the optimal performance of our auditing method under both $\mathcal{K}_t$ and $\mathcal{K}_w$. Our method achieves 100% accuracy on the Health-CareMagic dataset and 80% accuracy on the QA-Legal dataset when LLaMA3 is the teacher, far exceeding the 50% accuracy of random auditing. DLI also improves the accuracy by 45% over the best baseline of 55% to 100%. Results with GPT-4o-mini, Pythia, and Qwen2.5 as teacher mod-

| Model | Method | 0 | 1 | 2 | 3 | 4 | 5 | 6 | 7 | 8 | 9 | T |
|---|---|---|---|---|---|---|---|---|---|---|---|---|
| Student | MP | ✓ | ✗ | ✗ | ✗ | ✓ | ✗ | ✗ | ✓ | ✗ | ✓ | 4 |
| | PoS | ✓ | ✓ | ✓ | ✓ | ✓ | ✓ | ✓ | ✓ | ✓ | ✓ | 10 |
| | MPT | ✗ | ✓ | ✓ | ✓ | ✗ | ✓ | ✓ | ✓ | ✓ | ✓ | 8 |
| | Ours ($\mathcal{K}_t$) | ✓ | ✓ | ✓ | ✓ | ✓ | ✓ | ✓ | ✓ | ✓ | ✓ | 10 |
| | Ours ($\mathcal{K}_w$) | ✗ | ✓ | ✓ | ✓ | ✓ | ✓ | ✓ | ✓ | ✓ | ✓ | 9 |
| Non-Student | MP | ✗ | ✓ | ✓ | ✗ | ✓ | ✓ | ✓ | ✗ | ✓ | ✓ | 7 |
| | PoS | ✓ | ✗ | ✗ | ✗ | ✗ | ✗ | ✗ | ✗ | ✗ | ✗ | 1 |
| | MPT | ✗ | ✗ | ✗ | ✗ | ✗ | ✗ | ✗ | ✗ | ✗ | ✗ | 0 |
| | Ours ($\mathcal{K}_t$) | ✓ | ✓ | ✓ | ✓ | ✓ | ✓ | ✓ | ✓ | ✓ | ✓ | 10 |
| | Ours ($\mathcal{K}_w$) | ✓ | ✓ | ✓ | ✓ | ✓ | ✗ | ✓ | ✓ | ✗ | ✓ | 8 |

Table 3: Audit results on student and non-student models for LLaMA3 on the HealthCareMagic dataset. T denotes correct identifications (TP/TN).

els show the same pattern, confirming the versatility and effectiveness of our approach. Moreover, although setting $\mathcal{K}_w$ is more strict than setting $\mathcal{K}_t$, its performance is not always weaker. In fact, on the qa-legal dataset, when LLaMA3 is the teacher model, it performs even better than setting $\mathcal{K}_t$.

The detailed audit results are also reported in Table 3 and Table 4. In Table 3, 20 models are evaluated (10 distilled, 10 non-distilled). The baselines MP, Pos, and MPT are only able to correctly attribute 11, 11, and 8 models, respectively. In particular, the MP baseline, which uses model response embeddings, fails to capture the subtlety of attribution. The Pos and MPT methods perform poorly, as they simply attribute all models to a single class. In contrast, both the $\mathcal{K}_t$ and $\mathcal{K}_w$ settings are able to attribute substantially more models correctly. Similar results can be observed in Table 4, and additional results on the QA-Legal dataset and other teacher models are provided in Appendix A.

## 5.3 ABLATION STUDY

**Impact of Datasets.** To assess the sensitivity to knowledge of the distillation data, we evaluate three additional prompt datasets from different domains: (i) NQ-simplified Lukas22 (2023), a public set of Wikipedia question-answer (QA) pairs. (ii) SciQ for AI (2017), a domain-specific dataset consisting of 13,679 crowdsourced science exam questions across physics, chemistry, and biology. (iii) Finance causal lm (2023), financial news, and QA pairs for finance reasoning and comprehension. (iv) Multi-domain: we construct a mixed prompt dataset by sampling balanced subsets from five domains—*public*, *health*, *law*, *finance*, and *science*. This datasets simulates auditors who have no knowledge of the teacher's original distillation domain. Figure 3a shows that auditing accuracy achieves its best performance when the prompt dataset has a distribution similar to that of the distillation dataset. On the healthcare dataset, it can reach 100% accuracy. The multi-domain prompt set results in only a marginal accuracy drop of approximately 5%. With mismatched prompts such as SciQ, which differs substantially from the knowledge of the distillation model, performance declines yet remains about 80%. Similar trends are observed for the legal dataset, as shown in Figure 3b.

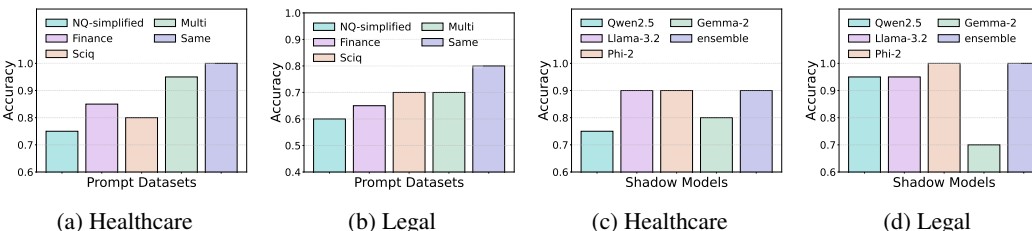

Figure 3: Impact of (a)-(b) prompt dataset choice and (c)-(d) shadow model choice on audit performance across healthcare and legal datasets.

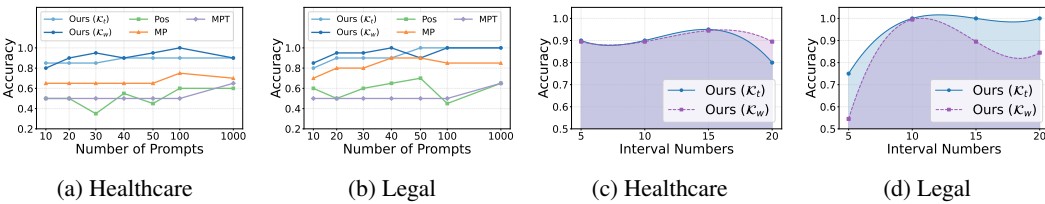

Figure 4: Impact of (a)-(b) vary the number of prompts and (c)-(d) vary the interval granularity on audit performance across healthcare and legal datasets.

**Impact of Shadow Models.** To evaluate the effectiveness of our method, which uses ensemble shadow models, we test it with four different models as shadow models: Qwen2.5-0.5B, LLaMA-3.2-1B, Phi-2, and Gemma-2-2B. Figure 3c shows that LLaMA-3.2 and Phi-2 perform best on the healthcare dataset, achieving about 90% accuracy, while Qwen2.5 performs slightly worse at around 75%. Therefore, to mitigate the impact of shadow model choice, we ensemble them, which achieves the best performance and greater robustness. A similar trend can also be observed in Figure 3d.

**Impact of Prompt Sizes.** To evaluate the impact of the number of prompts on our method, We vary the number of prompts from 10 to 1000 on two datasets and evaluate under $\mathcal{K}_t$ and $\mathcal{K}_w$ settings. As shown in Figure 4a, our method achieves over 80% accuracy even with only 10 prompts. Moreover, the $\mathcal{K}_t$ setting demonstrates greater stability, while $\mathcal{K}_w$ exhibits mild fluctuations yet remains strong. Increasing the prompt count improves stability but does

| Model | Method | 0 | 1 | 2 | 3 | 4 | 5 | 6 | 7 | 8 | 9 | T |
|---|---|---|---|---|---|---|---|---|---|---|---|---|
| Student | MP | ✗ | ✓ | ✗ | ✗ | ✗ | ✗ | ✓ | ✗ | ✓ | ✓ | 4 |
| | PoS | ✗ | ✓ | ✓ | ✗ | ✓ | ✓ | ✓ | ✓ | ✓ | ✓ | 8 |
| | MPT | ✓ | ✓ | ✗ | ✗ | ✗ | ✗ | ✓ | ✗ | ✗ | ✗ | 3 |
| | Ours ($\mathcal{K}_t$) | ✓ | ✓ | ✓ | ✓ | ✓ | ✓ | ✓ | ✗ | ✗ | ✓ | 8 |
| | Ours ($\mathcal{K}_w$) | ✓ | ✓ | ✓ | ✓ | ✓ | ✓ | ✓ | ✗ | ✗ | ✓ | 8 |
| Non-Student | MP | ✓ | ✓ | ✓ | ✓ | ✓ | ✓ | ✓ | ✓ | ✓ | ✓ | 10 |
| | PoS | ✓ | ✗ | ✗ | ✓ | ✓ | ✗ | ✗ | ✗ | ✓ | ✗ | 4 |
| | MPT | ✓ | ✓ | ✓ | ✓ | ✓ | ✓ | ✓ | ✓ | ✓ | ✓ | 10 |
| | Ours ($\mathcal{K}_t$) | ✓ | ✓ | ✓ | ✓ | ✓ | ✓ | ✓ | ✓ | ✓ | ✓ | 10 |
| | Ours ($\mathcal{K}_w$) | ✓ | ✓ | ✓ | ✓ | ✓ | ✓ | ✓ | ✓ | ✓ | ✓ | 10 |

Table 4: Audit results on student and non-student models for GPT-4o on the HealthCareMagic dataset.

not consistently raise attribution accuracy. Similar trends can also be observed among baselines. Consistent results are shown in Figure 4b, where prompts are drawn from the legal dataset.

**Impact of Interval Granularity.** We study how the number of probability intervals $m$ in the feature construction (Section 4.4) affects audit accuracy. We vary $m \in \{5, 10, 15, 20\}$ and evaluate on both datasets under both settings $\mathcal{K}_t$ and $\mathcal{K}_w$. Figure 4c-4d show that increasing $m$ from 5 yields consistent gains, indicating that finer discretization captures more informative variation in next token preferences. Beyond $m = 15$, performance degrades, as larger $m$ reduces per bin counts, which weakens the stability of the learned decision boundary and increases variance, especially in $\mathcal{K}_w$. Therefore, we select 10 intervals for the audit.

**Impact of Ensemble Sizes.** To assess the sensitivity of DLI to the shadow ensemble, we performed a systematic analysis by varying the number of shadow-distilled proxies from 1 to 4, chosen from Qwen2.5-0.5B, LLaMA-3.2-1B, Phi-2, and Gemma-2-2B. Figure 5a-5b shows that increasing the ensemble size and diversity can improve stability and robustness. While a single proxy achieves reasonable accuracy, its performance is not stable; adding more proxies substantially reduces variance across runs. The performance range (max–min) becomes much narrower as the ensemble size and diversity increase, indicating more stable and reliable audit decisions. Moreover, moving from 1 to 4 proxies yields measurable improvements in overall accuracy.

**Impact of Comparison Models.** To evaluate the impact of model-style token selection on our method, we experiment with four different comparison models: GPT2-large, Phi-2, Bloom-560M,

| Model | Method | Without Defense | | Back-Translation | | Cross-Teacher FT | | Inference-Time Rewrite | |
|---|---|---|---|---|---|---|---|---|---|
| | | Accuracy | F1-score | Accuracy | F1-score | Accuracy | F1-score | Accuracy | F1-score |
| LLaMA3 | $\mathcal{K}_t$ | 1.00 | 1.00 | 0.55↓ 45% | 0.18↓ 82% | 0.75↓ 25% | 0.67↓ 33% | 1.00↓ 0% | 1.00↓ 0% |
| | $\mathcal{K}_w$ | 0.85 | 0.86 | 0.50↓ 35% | 0.67↓ 19% | 0.70↓ 15% | 0.77↓ 9% | 0.60↓ 25% | 0.69↓ 17% |
| GPT-4o | $\mathcal{K}_t$ | 0.90 | 0.89 | 0.55↓ 35% | 0.18↓ 71% | 0.85↓ 5% | 0.82↓ 7% | 0.75↓ 15% | 0.67↓ 22% |
| | $\mathcal{K}_w$ | 0.90 | 0.89 | 0.55↓ 35% | 0.18↓ 71% | 0.85↓ 5% | 0.82↓ 7% | 0.90↓ 0% | 0.89↓ 0% |

Table 5: Performance of auditing methods under adaptive adversarial defenses.

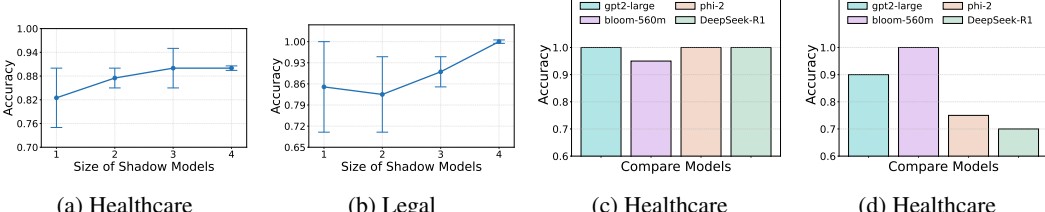

(a) Healthcare      (b) Legal      (c) Healthcare      (d) Healthcare

Figure 5: Impact of (a)-(b) vary the number of ensemble size and (c)-(d) vary the interval granularity on audit performance across healthcare and legal datasets.

and DeepSeek-R1-Distill-Qwen-1.5B (DeepSeek-R1). As shown in Figure 5c, when these models are used to select the model-style tokens of the LLaMA-3.1-8B teacher model (LLaMA3), the final audit performance indicates that almost all of them achieve good results. In contrast, Figure 5d shows that when auditing the teacher model GPT-4o-mini (GPT-4o), tokens selected using phi-2 and DeepSeek-R1 perform worse compared to those selected with GPT2-large and Bloom-560M. Therefore, for simplicity, we use GPT2-large as the comparison model for model-style token selection in the remainder of our experiments. Detailed guidelines and a probe-test procedure for selecting an appropriate comparison model are provided in Appendix A.5.

**Impact of Adaptive Adversary.** To evaluate the robustness of DLI under adaptive adversarial settings, we introduce three forms of output and training perturbations that reflect realistic deployer capabilities: (i) back-translation rewriting of the teacher-generated corpus before distillation, (ii) cross-teacher fine-tuning using data generated by a different teacher model within the same domain, and (iii) inference-time rewriting with a certain probability to mask stylistic cues without modifying model parameters. As shown in Table 5, these adversarial interventions degrade performance to varying degrees. Back-translation disrupts the original stylistic patterns and prevents the model from emitting teacher-specific tokens, thereby substantially weakening token-level stylistic signals. Cross-teacher fine-tuning causes moderate degradation, as conflicting stylistic patterns partially overwrite the original teacher's signature; the accuracy drops by approximately 5%–25% when LLaMA3 and GPT-4o serve as teacher models. Inference-time rewriting also induces moderate reduction, with accuracy decreases ranging from 0% to 25%, suggesting that even a small subset of preserved model-style tokens is sufficient for correct lineage auditing.

## 6 CONCLUSION AND LIMITATIONS

Our work formulates the problem of auditing distilled model provenance for LLMs, aimed at detecting potential licensing misuse. We propose a novel black-box auditing method, DLI, which enables model developers to determine whether their models have been distilled by other developers, thereby ensuring compliance with intellectual property protection policies. Extensive experiments demonstrate the effectiveness, robustness, and generalizability of our approach across both open-source and closed-source models. Despite these promising results, our study has two main limitations. First, due to compute and cost constraints, we use only four teachers (LLaMA3, GPT-4o-mini, Pythia-12b-v0 and Qwen2.5) to distill students; future work should test a broader set of LLMs to assess effects on model provenance auditing. Second, our current method is evaluated only on natural language processing models. Future work will aim to extend this approach to broader scenarios, such as multimodal settings, which would further broaden its applicability and impact.

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

| Shadow Model | Size | GPU | Training Time (hrs) |
|---|---|---|---|
| Qwen2.5 | 0.5B | 1×A100 | ∼0.24 |
| Llama-3.2 | 1B | 1×A100 | ∼0.28 |
| phi-2 | 2B | 1×A100 | ∼0.51 |
| gemma-2 | 2B | 1×A100 | ∼0.64 |

Table 6: Time required to train a single shadow model.

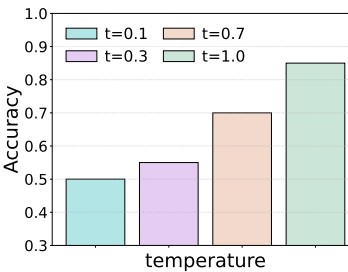 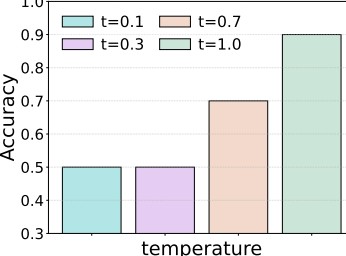

Figure 6: Impact of varying temperature on the audit performance on the teacher model LLaMA3 (left) and GPT-4o (right).

## A    APPENDIX

### A.1    LARGE LANGUAGE MODEL (LLM) USAGE DISCLOSURE

In accordance with the ICLR 2026 policy, we disclose the use of large language models in the preparation of this paper. We used LLMs (e.g., ChatGPT) solely for grammar checking and minor language polishing. No part of the research design, experiments, analysis, or substantive writing relied on LLMs.

### A.2    ADDITIONAL RESULTS

Here, we present the concrete audit results in Table 9, Table 10, Table 13, Table 14, Table 11 and Table 12. As shown in Table 9, among the 20 models, half are distilled and the other half are non-distilled. The baselines MP, Pos, and MPT are only able to correctly attribute 11, 10, and 6 models, respectively. In contrast, both the $\mathcal{K}_t$ and $\mathcal{K}_w$ settings are able to correctly attribute a larger number of models compared to the baselines. Similar results can be observed in Table 10, Table 13, Table 14, Table 11 and Table 12.

### A.3    EFFICIENCY STUDY

To evaluate the efficiency of our auditing framework, we analyze the computational overhead of constructing shadow models. Table 6 reports empirical training-time measurements for shadow models of different sizes (10k training samples, 1×A100 GPU). Even the largest 2B model requires only about 0.64 GPU hours, indicating that shadow training is lightweight. The overall cost also remains low when auditing many potential teachers simultaneously: the shadow models are relatively small (0.5B–2B) and train within minutes, and previously trained models can be reused for repeated or future audits, amortizing the cost. These factors make our auditing pipeline efficient and computationally practical at scale.

### A.4    MORE ABLATION STUDY

**Impact of Temperature.** We examine the robustness of stylistic tokens under varying temperature settings (0.1–1.0). Figure 6 shows that extremely low temperatures (0.1 and 0.3) suppress model-specific stylistic signals because sampling becomes nearly deterministic. This evaluation is conducted only under the **word-level access** ($K_w$) setting, which estimates token probabilities via repeated generation. The **token-level access** ($K_t$) setting cannot be evaluated in this manner, as it directly retrieves token probabilities from model logits and does not rely on sampled outputs. With moderate or high temperatures (0.7 and 1.0), stylistic differences re-emerge and auditor accuracy

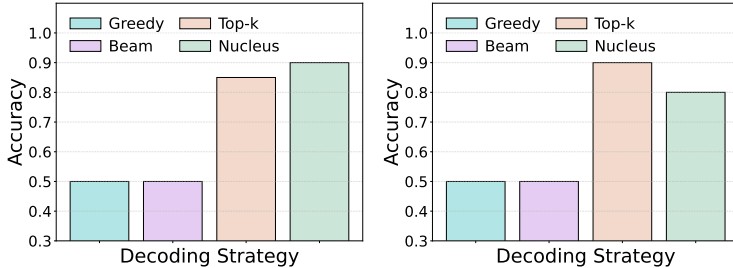

Figure 7: Impact of varying temperature on the audit performance on the teacher model LLaMA3 (left) and GPT-4o (right).

| Comparison Model ($M_c$) | LLaMA3 | GPT-4 |
|---|---|---|
| | score ↓ | score ↓ |
| gpt2-large | 0.376 | 0.106 |
| bloom-560m | 0.379 | **0.098** |
| phi-2 | 0.378 | 0.165 |
| DeepSeek-R1 | **0.321** | 0.140 |

Table 7: Probe test results for selecting comparison models ($M_c$). Lower scores indicate greater divergence from the teacher and better suitability for DLI.

increases, indicating that our method captures genuine model-dependent stylistic tendencies rather than artifacts of the generation procedure.

**Impact of Decoding Strategies.** We similarly evaluated different decoding strategies (greedy, beam search, top-k, and nucleus sampling) using the $K_w$ setting (Figure 7). Greedy and beam search collapse the distribution toward high-probability modes, reducing stylistic variability and auditor accuracy. Top-k and nucleus sampling preserve model-specific generative preferences while avoiding deterministic collapse, resulting in high accuracy. Again, $K_t$ is not applicable for this analysis because token probabilities are directly obtained from the model's logits without relying on generated sequences.

## A.5 SELECTION OF COMPARISON MODELS ($M_c$)

The choice of the comparison model ($M_c$) is crucial for effective auditing. A suitable $M_c$ should satisfy three properties: (i) **Sufficient stylistic divergence from the teacher.** $M_c$ should not belong to the same architectural family as the teacher, as overly similar models fail to highlight teacher-specific stylistic tokens. (ii) **Stable and well-calibrated token probabilities.** Reliable identification of teacher-specific differences requires $M_c$ to produce consistent probability distributions; highly noisy or unstable models reduce discriminative power. (iii) **Broad coverage of vocabulary space.** Models with collapsed or degenerate probability distributions provide little signal for auditing.

To assist auditors in selecting $M_c$ without prior knowledge of the teacher, we introduce a lightweight *probe test*. Using only 20 prompts, we compute, for each candidate $M_c$, the mean probability assigned to the teacher model's outputs:

$$\text{score}(M_c) = \mathbb{E}_x\big[P_{M_c}(y_{\text{teacher}} \mid x)\big].$$

Lower scores indicate stronger divergence from the teacher and thus a better ability to reveal teacher-specific stylistic tokens. Table 7 shows probe-test results for several candidate $M_c$ models when auditing LLaMA3 and GPT-4o. Even with only 20 prompts, the probe test reliably identifies strong candidates (e.g., `bloom-560m`) and flags unsuitable ones (e.g., `phi-2` for GPT-4o). This procedure provides both a practical guideline and empirical justification for selecting effective comparison models, with negligible computational cost.

## A.6 DISCUSSION

**Relationship to Membership Inference Attacks.** Although our approach superficially resembles shadow-model–based membership inference attacks (MIAs), the two are fundamentally different in

objective, threat model, and the nature of the statistical signal exploited. MIAs aim to determine whether a *sample* belonged to the training set of a target model, whereas our goal is to determine whether a *model* has been *distilled* from a specific teacher model. Consequently, MIAs perform **sample-level** inference, while DLI performs **model-level lineage inference**. Shadow models in MIAs approximate overfitting behavior of the target model, whereas in DLI, shadow student and non-student models approximate stylistic and distributional biases naturally transferred through distillation. Moreover, MIAs typically require knowledge about the public/private dataset split, while DLI operates with only black-box access and no assumptions about the training data.

**Relationship to Model Watermarking.** Model watermarking embeds artificial patterns or triggers into a model to prove ownership. In contrast, DLI does not rely on any embedded signal and leverages *natural* stylistic inheritance arising from distillation. Watermarking requires cooperation from the model owner and is inapplicable to proprietary teacher models. Additionally, watermark verification checks ownership, whereas DLI audits the distillation lineage of a model.

**Why DLI Works Better Than Existing Baselines.** Our framework benefits from two principled advantages. First, distillation transfers characteristic token-level behaviors from teacher to student models, which are naturally captured by our auditing method. Second, DLI employs a *contrastive* setup by comparing the suspected model against both shadow-student and shadow-non-student models, enabling the auditor to capture subtle stylistic signatures that absolute-similarity baselines fail to detect. This combination yields more reliable and accurate model-level lineage inference.

**Applicability of MIA or Watermarking Techniques.** Although conceptually related, existing techniques from MIA or model watermarking cannot be directly applied to our setting. MIAs focus on determining whether a *sample* was part of a model's training data, while our objective is to identify whether a *model* was distilled from a particular teacher. Watermarking requires pre-embedded signatures in the teacher model, which is incompatible with auditing proprietary, unmodified teacher models.

In summary, while DLI shares superficial similarities with MIAs and model watermarking, it addresses a fundamentally distinct problem: determining the lineage of a model at the *model level* rather than at the sample level. By leveraging natural stylistic inheritance and using a contrastive setup with shadow-student and shadow-non-student models, DLI captures subtle token-level signatures, achieving more accurate and robust lineage inference than existing baselines. This highlights both the novelty and practical applicability of our approach in auditing proprietary models under black-box conditions.

A.7    TERMINOLOGY AND NOTATION

Here we present the terminology and notation used throughout the Distillation Lineage Inspector (DLI) framework. Table 8 defines all key entities, datasets, and models referenced in the paper, providing a clear reference for readers.

| Term | Definition in Our Setting (DLI) |
|---|---|
| $T$ | Teacher model that generates the ground-truth training data for student models. |
| $M$ | Suspected deployed model whose lineage (student vs. non-student) is being inspected. |
| $M_{\text{audit}}$ | Auditor model trained to distinguish whether $M$ is distilled from $T$. |
| $M_s$ | Shadow *student* model distilled from $T$ (used for auditor training). |
| $M_n$ | Shadow *non-student* model not distilled from $T$ (used for auditor training). |
| $M_c$ | Comparison model; an auxiliary model used to measure relative similarity. |
| $\mathcal{X}$ | Query set used by the auditor to probe models. |
| $D_g$ | Generated dataset produced by querying models using $\mathcal{X}$. |
| $D_{\text{tokens}}$ | Model-style token lists extracted from each model's generation. |
| $D_{\text{audit}}$ | Final audit dataset consisting of per-sample features computed from token-level statistics. |

Table 8: Terminology and notation used in the Distillation Lineage Inspector (DLI) framework.

| Model | Method | 0 | 1 | 2 | 3 | 4 | 5 | 6 | 7 | 8 | 9 | T |
|---|---|---|---|---|---|---|---|---|---|---|---|---|
| Student | MP | ✓ | ✓ | ✓ | ✓ | ✗ | ✓ | ✓ | ✗ | ✗ | ✓ | 7 |
| | PoS | ✓ | ✓ | ✓ | ✓ | ✓ | ✓ | ✓ | ✓ | ✓ | ✓ | **10** |
| | MPT | ✗ | ✓ | ✓ | ✗ | ✓ | ✓ | ✓ | ✓ | ✗ | ✗ | 6 |
| | Ours ($\mathcal{K}_t$) | ✓ | ✓ | ✓ | ✗ | ✗ | ✓ | ✓ | ✗ | ✗ | ✓ | 6 |
| | Ours ($\mathcal{K}_w$) | ✗ | ✓ | ✓ | ✓ | ✓ | ✓ | ✓ | ✗ | ✓ | ✓ | 9 |
| Non-Student | MP | ✗ | ✓ | ✗ | ✗ | ✗ | ✓ | ✗ | ✓ | ✗ | ✓ | 4 |
| | PoS | ✗ | ✗ | ✗ | ✗ | ✗ | ✗ | ✗ | ✗ | ✗ | ✗ | 0 |
| | MPT | ✗ | ✗ | ✗ | ✗ | ✗ | ✗ | ✗ | ✗ | ✗ | ✗ | 0 |
| | Ours ($\mathcal{K}_t$) | ✓ | ✓ | ✓ | ✓ | ✓ | ✓ | ✓ | ✓ | ✓ | ✓ | **10** |
| | Ours ($\mathcal{K}_w$) | ✓ | ✓ | ✓ | ✓ | ✓ | ✓ | ✓ | ✓ | ✓ | ✓ | **10** |

Table 9: Audit results on student and non-student models for the teacher model LLaMA3 on the QA-Legal dataset.

| Model | Method | 0 | 1 | 2 | 3 | 4 | 5 | 6 | 7 | 8 | 9 | T |
|---|---|---|---|---|---|---|---|---|---|---|---|---|
| Student | MP | ✗ | ✓ | ✓ | ✓ | ✓ | ✗ | ✓ | ✗ | ✓ | ✓ | 7 |
| | PoS | ✓ | ✗ | ✓ | ✗ | ✗ | ✓ | ✓ | ✗ | ✗ | ✗ | 4 |
| | MPT | ✗ | ✓ | ✓ | ✗ | ✗ | ✓ | ✓ | ✓ | ✓ | ✓ | 7 |
| | Ours ($\mathcal{K}_t$) | ✓ | ✓ | ✓ | ✓ | ✓ | ✓ | ✓ | ✓ | ✓ | ✓ | **10** |
| | Ours ($\mathcal{K}_w$) | ✓ | ✓ | ✓ | ✓ | ✓ | ✓ | ✓ | ✓ | ✓ | ✓ | **10** |
| Non-Student | MP | ✓ | ✓ | ✓ | ✓ | ✓ | ✓ | ✓ | ✓ | ✓ | ✓ | 10 |
| | PoS | ✓ | ✓ | ✓ | ✓ | ✓ | ✗ | ✓ | ✓ | ✓ | ✓ | 9 |
| | MPT | ✓ | ✗ | ✓ | ✓ | ✓ | ✓ | ✓ | ✓ | ✗ | ✓ | 8 |
| | Ours ($\mathcal{K}_t$) | ✓ | ✓ | ✓ | ✓ | ✓ | ✓ | ✓ | ✓ | ✓ | ✓ | **10** |
| | Ours ($\mathcal{K}_w$) | ✓ | ✓ | ✓ | ✓ | ✓ | ✓ | ✓ | ✓ | ✓ | ✓ | **10** |

Table 10: Audit results on student and non-student models for the teacher model GPT-4o-mini on the QA-Legal dataset.

| Model | Method | 0 | 1 | 2 | 3 | 4 | 5 | 6 | 7 | 8 | 9 | T |
|---|---|---|---|---|---|---|---|---|---|---|---|---|
| Student | MP | ✗ | ✓ | ✓ | ✓ | ✓ | ✓ | ✗ | ✗ | ✓ | ✓ | 7 |
| | PoS | ✗ | ✗ | ✗ | ✗ | ✗ | ✗ | ✗ | ✗ | ✗ | ✗ | 0 |
| | MPT | ✗ | ✗ | ✗ | ✗ | ✗ | ✗ | ✗ | ✗ | ✗ | ✗ | 0 |
| | Ours ($\mathcal{K}_t$) | ✗ | ✓ | ✓ | ✓ | ✓ | ✓ | ✓ | ✓ | ✓ | ✓ | **9** |
| | Ours ($\mathcal{K}_w$) | ✓ | ✓ | ✗ | ✓ | ✗ | ✓ | ✓ | ✓ | ✓ | ✓ | **8** |
| Non-Student | MP | ✗ | ✓ | ✗ | ✓ | ✗ | ✓ | ✓ | ✓ | ✗ | ✓ | 6 |
| | PoS | ✓ | ✓ | ✓ | ✓ | ✓ | ✓ | ✓ | ✓ | ✓ | ✓ | 10 |
| | MPT | ✓ | ✗ | ✗ | ✗ | ✓ | ✗ | ✗ | ✓ | ✗ | ✗ | 3 |
| | Ours ($\mathcal{K}_t$) | ✓ | ✓ | ✓ | ✓ | ✓ | ✓ | ✓ | ✓ | ✓ | ✓ | **10** |
| | Ours ($\mathcal{K}_w$) | ✓ | ✓ | ✓ | ✓ | ✓ | ✓ | ✓ | ✓ | ✓ | ✓ | **10** |

Table 11: Audit results on student and non-student models for the teacher model pythia on the HealthCareMagic dataset.

| Model | Method | 0 | 1 | 2 | 3 | 4 | 5 | 6 | 7 | 8 | 9 | T |
|---|---|---|---|---|---|---|---|---|---|---|---|---|
| Student | MP | ✗ | ✗ | ✗ | ✗ | ✗ | ✓ | ✗ | ✓ | ✗ | ✗ | 2 |
| | PoS | ✗ | ✗ | ✗ | ✗ | ✗ | ✓ | ✗ | ✗ | ✓ | ✗ | 2 |
| | MPT | ✗ | ✗ | ✗ | ✗ | ✗ | ✗ | ✗ | ✗ | ✗ | ✗ | 0 |
| | Ours ($\mathcal{K}_t$) | ✓ | ✓ | ✓ | ✓ | ✓ | ✓ | ✓ | ✓ | ✓ | ✓ | **10** |
| | Ours ($\mathcal{K}_w$) | ✗ | ✓ | ✗ | ✓ | ✓ | ✓ | ✓ | ✓ | ✓ | ✓ | **8** |
| Non-Student | MP | ✗ | ✓ | ✓ | ✓ | ✓ | ✓ | ✓ | ✗ | ✗ | ✓ | 7 |
| | PoS | ✓ | ✗ | ✗ | ✗ | ✗ | ✓ | ✗ | ✗ | ✗ | ✗ | 2 |
| | MPT | ✗ | ✗ | ✓ | ✓ | ✓ | ✗ | ✗ | ✓ | ✗ | ✗ | 4 |
| | Ours ($\mathcal{K}_t$) | ✓ | ✓ | ✓ | ✓ | ✓ | ✓ | ✓ | ✓ | ✓ | ✓ | **10** |
| | Ours ($\mathcal{K}_w$) | ✓ | ✓ | ✓ | ✓ | ✓ | ✓ | ✓ | ✓ | ✓ | ✓ | **10** |

Table 12: Audit results on student and non-student models for the teacher model pythia on the QA-Legal dataset.

| Model | Method | 0 | 1 | 2 | 3 | 4 | 5 | 6 | 7 | 8 | 9 | T |
|---|---|---|---|---|---|---|---|---|---|---|---|---|
| Student | MP | ✗ | ✓ | ✓ | ✓ | ✓ | ✓ | ✓ | ✓ | ✓ | ✓ | 9 |
| | PoS | ✓ | ✓ | ✓ | ✓ | ✗ | ✓ | ✗ | ✗ | ✓ | ✓ | 7 |
| | MPT | ✗ | ✗ | ✗ | ✗ | ✗ | ✗ | ✓ | ✗ | ✗ | ✗ | 1 |
| | Ours ($\mathcal{K}_t$) | ✗ | ✓ | ✓ | ✓ | ✗ | ✓ | ✓ | ✗ | ✓ | ✓ | **7** |
| | Ours ($\mathcal{K}_w$) | ✗ | ✓ | ✓ | ✗ | ✗ | ✓ | ✓ | ✗ | ✗ | ✓ | **5** |
| Non-Student | MP | ✓ | ✗ | ✓ | ✓ | ✓ | ✗ | ✗ | ✗ | ✓ | ✗ | 5 |
| | PoS | ✗ | ✗ | ✗ | ✗ | ✗ | ✗ | ✗ | ✗ | ✗ | ✗ | 0 |
| | MPT | ✓ | ✓ | ✓ | ✓ | ✓ | ✓ | ✓ | ✓ | ✓ | ✓ | 10 |
| | Ours ($\mathcal{K}_t$) | ✓ | ✓ | ✓ | ✓ | ✓ | ✓ | ✓ | ✓ | ✓ | ✓ | **10** |
| | Ours ($\mathcal{K}_w$) | ✓ | ✓ | ✓ | ✓ | ✓ | ✓ | ✓ | ✓ | ✓ | ✓ | **10** |

Table 13: Audit results on student and non-student models for the teacher model Qwen2.5 on the HealthCareMagic dataset.

| Model | Method | 0 | 1 | 2 | 3 | 4 | 5 | 6 | 7 | 8 | 9 | T |
|---|---|---|---|---|---|---|---|---|---|---|---|---|
| Student | MP | ✓ | ✗ | ✗ | ✗ | ✗ | ✗ | ✗ | ✗ | ✗ | ✗ | 1 |
| | PoS | ✓ | ✓ | ✓ | ✓ | ✗ | ✓ | ✓ | ✓ | ✓ | ✓ | 9 |
| | MPT | ✗ | ✗ | ✗ | ✗ | ✗ | ✗ | ✗ | ✗ | ✗ | ✗ | 0 |
| | Ours ($\mathcal{K}_t$) | ✗ | ✓ | ✓ | ✓ | ✓ | ✓ | ✗ | ✓ | ✓ | ✓ | **9** |
| | Ours ($\mathcal{K}_w$) | ✗ | ✓ | ✗ | ✗ | ✗ | ✓ | ✓ | ✗ | ✗ | ✓ | **4** |
| Non-Student | MP | ✓ | ✓ | ✓ | ✓ | ✓ | ✓ | ✓ | ✓ | ✓ | ✓ | 10 |
| | PoS | ✗ | ✗ | ✗ | ✗ | ✗ | ✗ | ✗ | ✓ | ✗ | ✓ | 2 |
| | MPT | ✓ | ✓ | ✓ | ✓ | ✓ | ✓ | ✓ | ✓ | ✓ | ✓ | 10 |
| | Ours ($\mathcal{K}_t$) | ✓ | ✓ | ✓ | ✓ | ✓ | ✓ | ✓ | ✓ | ✓ | ✓ | **10** |
| | Ours ($\mathcal{K}_w$) | ✓ | ✓ | ✓ | ✓ | ✓ | ✓ | ✓ | ✓ | ✓ | ✓ | **10** |

Table 14: Audit results on student and non-student models for the teacher model Qwen2.5 on the QA-Legal dataset.