# OpenReview forum: "Distillation Lineage Inspector: Black-Box Auditing of Model Distillation in LLMs"
_ICLR.cc/2026/Conference — Submitted to ICLR 2026_

### Official Review · Reviewer_5PeB · 2025-10-31

**Soundness:** 2
**Presentation:** 2
**Contribution:** 2
**Rating:** 4
**Confidence:** 3

**Summary:**

This paper studies the problem of determining whether a large language model (LLM) has been used as a teacher to distill another model. In particular, the proposed approach resembles a (shadow-model based) membership inference setup: it assumes access to shadow models, including both distilled ($M_s$) and non-distilled ($M_n$) ones derived from the same teacher, as well as a generic comparison model ($M_c$). The method identifies model-style tokens, i.e., tokens that are assigned high probabilities by the teacher model $T$ but low probabilities by the comparison model $M_c$, and uses their likelihoods as discriminative features. A classifier is then trained on these features (from the shadow models) to distinguish distilled from non-distilled models. Experimental results demonstrate promising performance when using LLaMA3 or GPT-4o as teacher models across a diverse set of distilled and non-distilled candidates.

**Strengths:**

- This paper investigates an interesting and practically highly relevant topic.
- The proposed idea is reasonable and well-motivated.
- The experiments demonstrate intriguing results.

**Weaknesses:**

- The method description is somewhat unclear (see detailed questions below). In particular, the exact threat model (for both the proposed method and the baselines) is not explicitly defined, e.g., what level of access or knowledge does the attacker, model owner, or the proposed Distillation Lineage Inspector have? What actions can the attacker/model owner take (e.g., can the attacker “wash out” the feature indicators by further fine-tuning on related or unrelated data after the initial distillation)? what characteristics $M_c$ is expected to have (as from my understanding it should definitely be different from the teacher model)

- The experimental scale is relatively limited, making the results less convincing. Given the nature of the task, more extensive and diverse experiments are often necessary to support strong claims of verification and generalization.

**Questions:**

- Method description is somewhat confusing. From the pseudocode in Algorithm 1, the statement "append($D_{tokens},U$)" appears to be executed only once and does not occur within any iteration or loop. It is therefore unclear what the later lines 269–272: “for each token list $U_i \in D_{tokens}$ and "for each token $t_j \in U_i$ with position $p_j$ in $s_i$" are referring to.

- The naming conventions for “target data,” “dataset,” and “model” are also somewhat confusing (especially given their similarity—but not equivalence—to those used in membership inference literature). These terms could be made more explicit to avoid ambiguity.

- While the proposed approach resembles a shadow-model–based membership inference attack, the paper lacks deeper discussion on the principled difference or relationship between them (and potentially also the connection to model watermarking). Also, it would also be valuable to provide more intuition or analysis explaining why the proposed approach works better than existing baselines (e.g. what knowledge is particularly important for the inspector). Furthermore, it would be important to discuss whether techniques from the membership inference or model watermarking literature could be applied or extended to this setting.

- How is $M_n$ trained? Does it undergo standard fine-tuning on the local dataset?

- The origin of “for each input $x \in X$ is somehow unclear:  are these raw samples from the fine-tuning dataset? Moreover, referring to
$s$ as the “target sample” (e.g., in Algorithm 1) is potentially confusing in the context of privacy or membership inference literature. It would help to explicitly define both $x$ and  $s$ in the pseudocode and clearly distinguish their roles.

- From a metric perspective, it may be helpful to include results such as true positive rate (TPR) at low false positive rate (FPR) thresholds (e.g., 0.1%, 1%) to better illustrate detection performance.

---

> ### Author Response · Authors · 2025-11-26
>
> Q1. The method description is somewhat unclear (see detailed questions below). In particular, the exact threat model (for both the proposed method and the baselines) is not explicitly defined, e.g., what level of access or knowledge does the attacker, model owner, or the proposed Distillation Lineage Inspector have?
>
> A1. We thank the reviewer for raising this point. We clarify that the threat model is explicitly defined in Section~3.2 (Problem Formulation), and we summarize it here for completeness.
>
> Our setting involves three entities: (1) Model Developer}, (2) Suspected Deployer, and (3) Auditor (DLI). The audit objective is to determine whether the suspected model is a student distilled from a given teacher model.
> Model Developer: owns and trains a proprietary teacher model, which may be public or private, while retaining intellectual property (IP) rights.
> Suspected Deployer: serves a model via an API. The model may be a student-secretly distilled from a teacher model-raising concerns about unauthorized reuse.
> Auditor: operates under a black-box access assumption; it can query the suspected model and the teacher model, but has no access to the weights, training data, or internal activations. This defines the auditor's precise ``level of access and knowledge.''Notably, there is no ``attacker'' in our formulation; we adopt standard passive auditing assumptions consistent with prior provenance work.
>
> For baselines, we follow their recommended black-box configurations to ensure fairness. All methods, including our proposed DLI and the baselines, are evaluated under the same threat model.
>
> Q2. What actions can the attacker/model owner take (e.g., can the attacker wash out the feature indicators by further fine-tuning on related or unrelated data after the initial distillation)?
>
> A2. We thank the reviewer for raising this critical point. To evaluate whether a suspected deployer can wash out teacher-specific stylistic indicators, we consider the cross-teacher fine-tuning scenario, where the distilled model is further fine-tuned on data generated by a different teacher model within the same domain. As shown in Table 5, this intervention causes moderate degradation of audit performance, indicating that some stylistic signals are partially overwritten. However, the auditor remains effective; even with conflicting stylistic patterns introduced by fine-tuning, DLI still achieves 85\% accuracy. This demonstrates that while additional fine-tuning can attenuate model-style features, it cannot fully erase the inherent stylistic dependencies inherited during distillation.
>
> Q3. what characteristics $M_c$ is expected to have (as from my understanding it should definitely be different from the teacher model.
>
> A3. We thank the reviewer for this important question. A good comparison model should satisfy the following properties: (1) Sufficient stylistic divergence from the teacher. $M_c$ should not share the same architectural family as the teacher. If $M_c$ is too similar to the teacher, the relative logits will not highlight teacher-specific stylistic tokens. (2) Stable and well-calibrated token probabilities. The goal is to identify where the teacher and $M_c$ systematically disagree. Models with unstable or highly noisy probability distributions make this comparison less reliable. (3) Broad coverage of vocabulary space.  $M_c$ should produce non-degenerate probabilities across the candidate tokens; overly small or extremely weak models tend to collapse probability mass and reduce discriminative power.
>
> To allow auditors to select an appropriate $M_c$ without prior knowledge of the teacher, we introduce a simple probe test procedure. Using only 20 prompts, we compute, for each candidate $M_c$, the mean probability that $M_c$ assigns to the teacher model's outputs.  score($M_c$) = $E_x$ [ $P_{M_c}$ ($y_{teacher}$ $\mid x)$], A lower score indicates stronger divergence between $M_c$ and the teacher; therefore, it better reveals teacher-specific stylistic tokens. This procedure reliably identifies effective comparison models with negligible computational costs.
>
> Table 7 shows probe-test results for several candidate $M_c$ models when auditing LLaMA3 and GPT-4o. Even with only 20 prompts, the probe test correctly flags unsuitable choices (e.g., phi-2 for GPT-4o) and identifies strong candidates (e.g., bloom-560m). This probe test thus provides both a practical guideline and an empirical justification for what makes a ``good'' comparison model. We have added these  in the revised manuscript.

---

> > ### Author Response · Authors · 2025-11-26
> >
> > Q4. The experimental scale is relatively limited, making the results less convincing. Given the nature of the task, more extensive and diverse experiments are often necessary to support strong claims of verification and generalization.
> >
> > A4. We thank the reviewer for pointing this out. To evaluate the generalizability of the "model-style token" signature beyond LLaMA-3-8B and GPT-4o-mini, we conducted additional experiments using two more teacher models: Pythia-12B and Qwen2.5-7B. These models differ in both architecture and pretraining corpus, providing a stronger test of cross-family generalization. We have included these new experiments and their results in the revised manuscript.
> >
> > As shown in Table 2 of the revised manuscript, the "model-style token" signature consistently captures teacher-specific stylistic signals across these additional architectures. Our method achieves 100\% accuracy when Pythia-12B is the teacher model and 95\% accuracy when Qwen2.5-7B is the teacher, confirming that our approach generalizes well to different model families and scales.
> >
> > Q5. Method description is somewhat confusing. From the pseudocode in Algorithm~1, the statement append($D_{tokens}$, $U$) appears to be executed only once and does not occur within any iteration or loop. It is therefore unclear what the later lines 269--272, namely for each token list $U_i \in D_{tokens}$ and or each token $t_j \in U_i$ with position $p_j$ in $s_i$,  are referring to.
> >
> > A5. We thank the reviewer for pointing out this ambiguity. After re-examining Algorithm~1, we agree that the pseudocode did not explicitly show the outer loop over all input samples, which made it unclear how multiple token lists $U_i$ were accumulated into $D_{tokens}$.
> >
> > In our method, the input to the algorithm is the complete set of samples from the generated dataset $D_g$ = {$s_1, s_2, \dots, s_N$}.  For each sample $s_i$, we compute its corresponding model-style token list $U_i$, and then append $U_i$ to  $D_{tokens}$ = {$U_1, U_2, \dots, U_N$}. The original pseudocode only showed the inner logic for extracting $U_i$ from a single sample, which led to the confusion observed by the reviewer. To address this, we have updated Algorithm~1 in the revised manuscript to explicitly include the outer loop over all samples in $D_g$ and to clearly indicate that each sample produces its own token list that is stored in $D_{tokens}$.
> >
> > Q6. The naming conventions for target data, dataset, and model are somewhat confusing (especially given their similarity---but not equivalence---to those used in the membership inference literature). These terms could be defined more explicitly to avoid ambiguity.
> >
> > A6. We thank the reviewer for pointing out the ambiguity in our naming conventions. To avoid confusion, especially regarding terminology commonly used in the membership-inference literature, we have revised the manuscript to define all key terms explicitly. We also added Table 8 to distinguish the concepts used in our auditing setting clearly. These definitions clarify the role of each term. We have incorporated these explicit definitions and the terminology table into the revised manuscript.
> >
> > Q7. How is $M_n$ trained? Does it undergo standard fine-tuning on the local dataset?
> >
> > A7. $M_n$ refers to a non-student model that serves as a negative example for the auditor. Importantly, $M_n$ is pre-trained by the model developer and is drawn from a different model family than that of the teacher. It does not undergo any additional fine-tuning or adaptation on the local dataset. We use $M_n$ as-is to simulate models unrelated to the teacher, providing features that clearly distinguish it from student models.
> >
> > Q8. The origin of for each input $x \in X$ is somewhat unclear: are these raw samples from the fine-tuning dataset? Moreover, referring to $s$ as the target sample (e.g., in Algorithm~1) is potentially confusing in the context of privacy or membership inference literature. It would be helpful to explicitly define both $x$ and $s$ in the pseudocode and clearly distinguish their roles.
> >
> > A8. We thank the reviewer for pointing this out. In our setting, the set $\mathcal{X}$ is not drawn from any fine-tuning dataset; rather, it is the auditor's own query set, as described in Sec 3.2 (Auditor Capability).  Each sample $x \in \mathcal{X}$ is first split into two parts: the first part is used as a prompt to query the teacher model, which then generates a new sample $s$ corresponding to $x$.  In Algorithm~1, $x$ therefore represents a query sample from the auditor's set, while $s$ is the generated sample produced by the teacher (forming the generated dataset $D_g$), which is then used with the comparison model to identify model-style tokens.  We have updated the pseudocode and text to explicitly define $x$ and $s$, clearly distinguishing their roles and avoiding potential confusion with terminology from privacy or membership inference literature.

---

> > > ### Author Response · Authors · 2025-11-26
> > >
> > > Q9. While the proposed approach resembles a shadow-model--based membership inference attack, the paper lacks deeper discussion on the principled difference or relationship between them (and potentially also the connection to model watermarking). Moreover, it would be valuable to provide more intuition or analysis explaining why the proposed approach works better than existing baselines (e.g., what knowledge is particularly important for the inspector). Furthermore, it would be important to discuss whether techniques from the membership inference or model watermarking literature could be applied or extended to this setting.
> > >
> > > A9. We thank the reviewer for this insightful comment. Although our approach superficially resembles a shadow-model-based membership inference attack (MIA), the two are fundamentally different in their objectives, threat models, and the nature of the statistical signal they exploit. We have added a dedicated discussion section in the revised paper.
> > >
> > > Difference from Membership Inference Attacks. MIA aims to determine whether a sample belongs to the training set of a target model, whereas our goal is to determine whether a model has been distilled from a specific teacher model. Thus, MIAs perform sample-level inference, while DLI performs model-level lineage inference.  Shadow models in MIAs approximate overfitting behavior, whereas in DLI, shadow student and non-student models approximate the stylistic and distributional biases that are naturally transferred through distillation.  Further, MIAs typically require assumptions about the public/private dataset split of the target model, while DLI operates with only black-box access and without any assumptions about the training data.
> > >
> > > Relationship to Model Watermarking. Model watermarking embeds artificial patterns or triggers into a model to prove ownership. In contrast, DLI does not rely on any embedded signal and instead leverages natural stylistic inheritance that arises from distillation.
> > > Watermarking requires the cooperation of the model owner and does not apply to proprietary teachers (e.g., OpenAI, Google). Additionally, watermark verification checks ownership, whereas DLI checks distillation lineage. We have already talked about this in Section 2.
> > >
> > > Why DLI Works Better Than Existing Baselines. Our method benefits from two principled advantages: (i) distillation transfers characteristic token-level behaviors; and  (ii) DLI uses a contrastive setup by comparing the suspected model against both shadow-student and shadow-non-student models, enabling the auditor to capture subtle stylistic signatures that absolute-similarity baselines fail to detect.
> > >
> > > Applicability of MIA or Watermarking Techniques. Although conceptually related, techniques from MIA or model watermarking cannot be directly applied to our setting.  First, the goal of MIAs is fundamentally different from ours: MIAs focus on determining whether a sample was part of a model's training data, whereas our objective is to determine whether a model was distilled from a specific teacher.  Second, watermarking techniques require pre-embedded signatures in the teacher model, which is incompatible with our assumption that the teacher is an unmodified proprietary model.
> > >
> > > Q10. From a metric perspective, it may be helpful to include results such as true positive rate (TPR) at low false positive rate (FPR) thresholds (e.g., 0.1\%, 1\%) to better illustrate detection performance.
> > >
> > > A10. We appreciate the reviewer's suggestion to report metrics such as TPR at low FPR thresholds (e.g., 0.1\% or 1\%). After careful consideration, we decided not to include these metrics because they are poorly aligned with our evaluation setup or the scale of our test set.
> > >
> > > Our evaluation involves a total of 20 models (10 student models and 10 non-student models). Measuring the false positive rate (FPR) at 0.1\% or 1\% is statistically not meaningful because a 1\% FPR corresponds to only 0.1 negative samples, and a 0.1\% FPR corresponds to 0.01 samples, which cannot be reliably estimated given the small number of test models.
> > >
> > > Instead, we follow prior work and report accuracy, which is the most stable and well-defined metric given the discrete nature of the model-level classification task. Accuracy fully reflects auditor performance in our setting, where each model constitutes a single test instance.

---

### Official Review · Reviewer_m8S8 · 2025-11-01

**Soundness:** 3
**Presentation:** 3
**Contribution:** 3
**Rating:** 6
**Confidence:** 4

**Summary:**

This paper addresses the intellectual property and legal concerns of unauthorized model distillation, where a "student" model is trained using knowledge from a proprietary "teacher" model. The authors propose the Distillation Lineage Inspector (DLI), a novel framework to audit a model's provenance in a black-box setting. DLI works by training a binary classifier (an "auditor model") to distinguish between distilled and non-distilled models. To do this, the auditor first creates "shadow-distilled proxies" by training models with the teacher model (as positive examples) and without it (as negative examples). The framework's key insight is to identify "model-style tokens", tokens that the teacher model has a high probability of generating but a generic "comparison model" finds unlikely. The auditor model is then trained on feature vectors, which are histograms of the probabilities that the shadow models assign to these specific style tokens. To test a suspicious model, DLI generates its probability histogram for these tokens and feeds it to the trained auditor for a final decision. The paper evaluates DLI under two conditions: Token-Level Access (Kt), with access to token probabilities, and the more restrictive Word-Level Access (Kw), with access to only the final text. Experiments show DLI achieves high accuracy (e.g., 100% in Kt and 85% in Kw on the HealthCareMagic dataset) and significantly outperforms baselines by up to 45%.

**Strengths:**

- This paper tackles a timely and critical problem: the need for auditing AI models to protect against intellectual property theft via unauthorized distillation.
- A significant contribution is reframing the problem from "closed-world identification" (choosing from a list of known teachers) to "open-world verification" (confirming lineage from one specific teacher), which is a much more realistic auditing scenario.
- The DLI framework is model-agnostic and designed for purely black-box settings, making it applicable to auditing closed-source commercial APIs, not just open-source models.
- The method demonstrates remarkable robustness. It performs well even under the highly constrained Word-Level Access (Kw) (word-level only) setting by estimating probabilities via sampling, which is a practical solution for a real-world auditor.
- The empirical results are very strong. DLI achieves near-perfect accuracy in the Token-Level Access (Kt) setting and a 45% absolute accuracy improvement over the best-performing baseline on the healthcare dataset. It is also highly data-efficient, achieving over 80% accuracy with just 10 prompts.

**Weaknesses:**

- The framework's effectiveness is only validated using two teacher models (Llama-3-8B and GPT-4o-mini), a limitation the authors acknowledge. It is unclear how well the "model-style token" signature generalizes to other model families and architectures.
- The crucial step of "Identifying Model-Style Tokens" depends on a generic comparison model (Mc). The ablation in Appendix A.3 (Figure 5) shows that the choice of Mc significantly impacts performance; for example, auditing GPT-4o failed with some comparison models. This suggests that finding an effective Mc is a non-trivial prerequisite for the auditor.
- The DLI framework is complex, requiring the auditor to execute a multi-stage pipeline: (1) distill an ensemble of shadow proxies , (2) select a suitable comparison model , (3) generate a labeled dataset , and (4) train a final classifier using AutoML. This complexity may be a barrier to its practical adoption.
- The baseline methods (PoS, MPT) perform exceptionally poorly , with MPT and PoS scoring 0% on correctly identifying non-students in some experiments (e.g., Table 5). This may suggest the baselines, which were not designed for this "open-world verification" task, are not strong points of comparison, making DLI's dominance appear larger.
- The paper's limitations note that the method is designed only for NLP models and cannot be extended to multimodal settings.

**Questions:**

- The choice of the "comparison model" (Mc) seems critical, as a poor choice (like phi-2 for auditing GPT-4o) leads to bad performance. How should an auditor practically select an effective Mc without prior knowledge? What properties make a comparison model "good" or "bad" for this task?
- In the Word-Level Access (Kw) (word-level) setting, probabilities are estimated using only N=5 samples. This results in very coarse probability estimates (e.g., 0.0, 0.2, 0.4, etc.). How does this coarse-grained estimation interact with the feature-encoding histogram, which uses 10 bins? Is this low N the primary reason for the performance drop from Token-Level Access (Kt)?
- The DLI framework requires the auditor to train an ensemble of shadow models. How sensitive is the final audit accuracy to the size and diversity of this shadow ensemble? Would the framework still work if the auditor only trained a single shadow-distilled proxy?
- The baselines (PoS, MPT) performed very poorly, especially at identifying non-students (e.g., 0/10 correct in Table 5). Is this because these methods are fundamentally unsuited for open-world verification and are only designed for closed-world identification, thus making them "strawman" comparisons?
- Could the DLI framework be inverted to solve the closed-world identification problem? For example, could an auditor pre-train 10 different auditors, each for a different major teacher model, and run a suspicious model against all 10 to determine which teacher it was most likely distilled from?

---

> ### Author Response · Authors · 2025-11-26
>
> Q1. The framework's effectiveness is only validated using two teacher models (Llama-3-8B and GPT-4o-mini), a limitation the authors acknowledge. It is unclear how well the "model-style token" signature generalizes to other model families and architectures.
>
> A1. We thank the reviewer for pointing this out. To evaluate the generalizability of the "model-style token" signature beyond LLaMA-3-8B and GPT-4o-mini, we conducted additional experiments using two more teacher models: Pythia-12B and Qwen2.5-7B. These models differ in both architecture and pretraining corpus, providing a stronger test of cross-family generalization. We have included these new experiments and their results in the revised manuscript.
>
> As shown in Table 2 of the revised manuscript, the "model-style token" signature consistently captures teacher-specific stylistic signals across these additional architectures. Our method achieves 100\% accuracy when Pythia-12B is the teacher model and 95\% accuracy when Qwen2.5-7B is the teacher, confirming that our approach generalizes well to different model families and scales.
>
> Q2. The crucial step of "Identifying Model-Style Tokens" depends on a generic comparison model (Mc). The ablation in Appendix A.3 (Figure 5) shows that the choice of Mc significantly impacts performance; for example, auditing GPT-4o failed with some comparison models. This suggests that finding an effective Mc is a non-trivial prerequisite for the auditor.
>
> The choice of the "comparison model" (Mc) seems critical, as a poor choice (like phi-2 for auditing GPT-4o) leads to bad performance. How should an auditor practically select an effective Mc without prior knowledge? What properties make a comparison model "good" or "bad" for this task?
>
> A2. We thank the reviewer for this important question. Indeed, the choice of the comparison model ($M_c$) is crucial.
>
> A good comparison model should satisfy the following properties: (1) Sufficient stylistic divergence from the teacher. $M_c$ should \emph{not} share the same architectural family as the teacher. If $M_c$ is too similar to the teacher, the relative logits will not highlight teacher-specific stylistic tokens. (2) Stable and well-calibrated token probabilities. The goal is to identify where the teacher and $M_c$ systematically disagree. Models with unstable or highly noisy probability distributions make this comparison less reliable.
> (3) Broad coverage of vocabulary space.  $M_c$ should produce non-degenerate probabilities across the candidate tokens; overly small or extremely weak models tend to collapse probability mass and reduce discriminative power.
>
> To allow auditors to select an appropriate $M_c$ without prior knowledge of the teacher, we introduce a simple probe test procedure. Using only 20 prompts, we compute, for each candidate $M_c$, the mean probability that $M_c$ assigns to the teacher model's outputs.  score($M_c$) = $E_x$ [ $P_{M_c}$ ($y_{teacher}$ $\mid x$], A lower score indicates stronger divergence between $M_c$ and the teacher; therefore, it better reveals teacher-specific stylistic tokens. This procedure reliably identifies effective comparison models with negligible computational costs.
>
> Table 7 shows probe-test results for several candidate $M_c$ models when auditing LLaMA3 and GPT-4o. Even with only 20 prompts, the probe test correctly flags unsuitable choices (e.g., phi-2 for GPT-4o) and identifies strong candidates (e.g., bloom-560m). This probe test thus provides both a practical guideline and an empirical justification for what makes a ``good'' comparison model. We have added these  in the revised manuscript.
>
> Q3. The DLI framework is complex, requiring the auditor to execute a multi-stage pipeline: (1) distill an ensemble of shadow proxies , (2) select a suitable comparison model , (3) generate a labeled dataset , and (4) train a final classifier using AutoML. This complexity may be a barrier to its practical adoption.
>
> A3. We thank the reviewer for highlighting this concern. Despite the multi-stage structure, the practical overhead is modest. Table 6 reports empirical training times. Even the largest 2B-parameter shadow model trains in only 0.64 GPU hours. Our ensemble uses four such models, but in constrained settings, we find that a single shadow model (e.g., Qwen2.5-0.5B) still achieves strong performance, enabling further simplification. Comparison-model selection via our probe test requires only 20 prompts and has negligible cost (typically under 1 minute). Labeled-dataset generation uses 1,000 prompts and completes in about 0.25 GPU hours on a single A100 GPU for one shadow model. Finally, the AutoML classifier runs entirely on CPU and finishes in under one minute.
>
> Thus, although DLI is a multi-stage pipeline, the end-to-end compute cost is low-well within the budget of a small GPU cluster or even a single GPU. We have added this analysis in the revised manuscript to demonstrate that DLI is practical and lightweight.

---

> > ### Author Response · Authors · 2025-11-26
> >
> > Q4. The baseline methods (PoS, MPT) perform exceptionally poorly , with MPT and PoS scoring 0\% on correctly identifying non-students in some experiments (e.g., Table 5). This may suggest the baselines, which were not designed for this "open-world verification" task, are not strong points of comparison, making DLI's dominance appear larger. The baselines (PoS, MPT) performed very poorly, especially at identifying non-students (e.g., 0/10 correct in Table 5). Is this because these methods are fundamentally unsuited for open-world verification and are only designed for closed-world identification, thus making them "strawman" comparisons?
> >
> > A4. We thank the reviewer for raising this point. The reviewer is correct that PoS was originally developed for closed-world model identification, where the auditor assumes that the true teacher is always contained within a fixed candidate set. MPT, in contrast, was developed for open-world verification, using a control model set to determine whether a queried model is fine-tuned from a particular pretrained model.
> >
> > Therefore, these methods are not 'strawman' comparisons. They represent the most recent and most relevant model-provenance baselines available in the literature. Prior work on PoS and MPT explicitly examines fingerprints of distilled or fine-tuned models, making them the closest existing approaches that could, in principle, be adapted to our setting. For fairness, we evaluated them exactly under their recommended configurations. We also note that Table~5 reports results for only one domain (QA-Legal). Across other datasets and teacher models, PoS and MPT show slightly improved performance, but they still consistently fail to distinguish student from non-student models. This reflects a fundamental limitation of these baselines rather than an issue with our experimental design.
> >
> > Our contribution is precisely to address this gap: DLI is the first method designed explicitly for open-world verification, in which the auditor must reject unmatched models. The poor performance of PoS and MPT empirically demonstrates the need for a new framework rather than an inappropriate baseline selection.
> >
> > Q5. The paper's limitations note that the method is designed only for NLP models and cannot be extended to multimodal settings.
> >
> > A5. We appreciate the reviewer's comments. We would like to clarify that our limitation statement does not claim that DLI is inherently restricted to NLP-only settings. Rather, our current study evaluates and demonstrates the framework only on text-generation models; therefore, we deliberately avoid concluding on multimodal systems.
> >
> > In principle, the core mechanism of DLI-tracking distributional shifts in token-level predictive behavior can be extended to multimodal architectures. However, doing so would require modality-specific feature definitions (e.g., vision-token or audio-token predictive distributions). We therefore frame multimodal extension as future work, not because it is impossible, but because it requires additional engineering and evaluation to ensure robustness across modalities.
> >
> > Q6. In the Word-Level Access (Kw) (word-level) setting, probabilities are estimated using only N=5 samples. This results in very coarse probability estimates (e.g., 0.0, 0.2, 0.4, etc.). How does this coarse-grained estimation interact with the feature-encoding histogram, which uses 10 bins? Is this low N the primary reason for the performance drop from Token-Level Access (Kt)?
> >
> > A6. We thank the reviewer for this careful question. In the Word-Level Access ($K_w$) setting, we estimate each token's probability using $N=5$ samples, which produce coarse-grained values (0.0, 0.2, 0.4, etc.). This coarse granularity slightly reduces the resolution of the 10-bin histogram, as multiple probabilities are mapped to the same bin, which could blur subtle distinctions between teacher and non-teacher distributions.
> >
> > However, in practice, we observe that $K_w$ achieves comparable performance to $K_t$ in most cases (and sometimes slightly better, e.g., LLaMA3 on QA-Legal), indicating that even coarse-grained probabilities preserve the essential teacher-specific signals. Increasing $N$ would indeed improve the histogram resolution and better approximate token-level probabilities, but at a significantly higher query cost and latency. We selected $N=5$ as a practical trade-off that balances audit accuracy with efficiency, showing that meaningful teacher-specific features can be extracted even from a small number of samples.

---

> > > ### Author Response · Authors · 2025-11-26
> > >
> > > Q7. The DLI framework requires the auditor to train an ensemble of shadow models. How sensitive is the final audit accuracy to the size and diversity of this shadow ensemble? Would the framework still work if the auditor only trained a single shadow-distilled proxy?
> > >
> > > A7. We thank the reviewer for this insightful question. To assess the sensitivity of DLI to the shadow ensemble, we performed a systematic analysis by varying the number of shadow-distilled proxies from 1 to 4, chosen from Qwen2.5-0.5B, LLaMA-3.2-1B, Phi-2, and Gemma-2-2B.
> > >
> > > Figure 5(a)-(b) shows that increasing the ensemble size and diversity can improve stability and robustness. While a single proxy achieves reasonable accuracy, its performance is not stable; adding more proxies substantially reduces variance across runs. The performance range (max-min) becomes much narrower as the ensemble size and diversity increase, indicating more stable and reliable audit decisions. Moreover, moving from 1 to 4 proxies yields measurable improvements in overall accuracy.
> > >
> > > We have included these ablation experiments in our revised manuscript.
> > >
> > > Q8. Could the DLI framework be inverted to solve the closed-world identification problem? For example, could an auditor pre-train 10 different auditors, each for a different major teacher model, and run a suspicious model against all 10 to determine which teacher it was most likely distilled from?
> > >
> > > A8. Yes, the DLI framework can indeed be adapted to address closed-world identification tasks. In such a setting, an auditor could pre-train multiple auditor models, each specialized for a major teacher model, and then evaluate a suspicious model against all of them. The auditor would select the teacher model whose corresponding auditor produces the strongest stylistic alignment with the suspicious model. Conceptually, this approach leverages the same core ideas of shadow-model distillation and stylistic-token identification; the main difference is that the closed-world setting restricts the candidate set of teachers, allowing the auditor to perform a direct comparison rather than open-world verification. We consider this a natural extension of DLI and note that the framework is flexible enough to support both open- and closed-world auditing scenarios.

---

### Official Review · Reviewer_YXqc · 2025-11-02

**Soundness:** 2
**Presentation:** 2
**Contribution:** 2
**Rating:** 4
**Confidence:** 4

**Summary:**

This paper studies the problem of auditing whether a suspicious model has been illicitly distilled from a proprietary large language model (LLM), even in strict black box API settings. To address this, the authors propose Distillation Lineage Inspector (DLI), which is a model-agnostic auditing framework. DLI works under both token level access (log-probabilities available) and word level access (content only). Experiments on healthcare and legal QA datasets show that DLI significantly outperforms existing provenance methods.

**Strengths:**

1. The problem considered in this paper is very timely and it has significant real-world relevance.
2. The proposed DLI requires no access to model internals, training data, or architecture... this is super crucial for practical auditing purposes..
3. The authors provided a very strong empirical performance by using multiple datasets.

**Weaknesses:**

1. The proposed framework relies on constructing shadow student models for each teacher. This is compute intensive task and may not scale to many suspect teachers simultaneously..
2. Performance drops meaningfully when prompts are distribution mismatched. This may limit applicability where the original distillation domain is unknown.
3. Authors did not carry out any analysis on adaptive adversaries.. which is an important concern for practical usage of the proposed approach.

**Questions:**

Please address the above issues highlighted in the weakness section.

Furthermore, I have the following comments as well:
(a) Considering adaptive attackers.. they may mask stylistic token patterns.. this calls out for an ablation on adversarial output perturbation.
(b) When auditing many candidate teachers, do auditors need separate shadow proxies for each? Provide computational complexity estimates.
(c) How stable are these stylistic tokens across: temperature changes, decoding strategies (greedy, nucleus, beam), etc.?
(d) Some unrelated models may coincidentally share stylistic preferences. When does DLI misfire?

---

> ### Author Response · Authors · 2025-11-26
>
> Q1. The proposed framework relies on constructing shadow student models for each teacher. This is compute intensive task and may not scale to many suspect teachers simultaneously. When auditing many candidate teachers, do auditors need separate shadow proxies for each? Provide computational complexity estimates.
>
> A1. We thank the reviewer for raising this important point. Indeed, our framework requires training shadow student models for each candidate teacher model. Because a shadow student model is trained to imitate the stylistic and probabilistic behaviors of a specific teacher's student model, each teacher model necessarily requires its own shadow student proxy. This is fundamental to the design of our auditing framework and cannot be shared across different teachers.
>
> Table 6 (in the appendix) provides empirical training-time measurements for shadow models of different sizes. Even the largest 2B shadow model trains in only about 0.64 GPU hours. The cost remains manageable even when auditing many suspect teachers simultaneously for three reasons: (1) the shadow models are small (0.5B-2B) and can be trained within minutes; (2) once trained, shadow students can be reused for repeated or future audits, amortizing the cost; and (3) our results in Figure 5(a)(b) show that even 1-2 shadow models capture most teacher-specific stylistic signals, further reducing the required compute. We have added these clarifications and computational estimates to the revised manuscript.
>
> Q2. Performance drops meaningfully when prompts are distribution mismatched. This may limit applicability where the original distillation domain is unknown.
>
> A2. We thank the reviewer for highlighting this concern. We agree that prompt-distribution mismatch can, in principle, affect audit performance when the original distillation domain is unknown. To evaluate the robustness of our method under realistic domain uncertainty, we conducted an additional experiment using a multi-domain prompt set. Specifically, we constructed a prompt pool by sampling queries from five distinct domains: public, health, law, finance, and science. For each domain, we extracted a balanced subset of prompts and combined them into a unified query set, simulating scenarios in which auditors have no knowledge of the teacher's original distillation domain.
>
> Our auditing framework remains highly robust under this setting. As shown in Figure 3(a)-(b), while domain-matched prompts yield the best performance, the multi-domain prompt set results in only a marginal accuracy drop of approximately 5-10\%. Importantly, it still consistently outperforms the use of prompts from a completely unrelated single domain. These findings indicate that the method remains practical even when the true distillation domain is unknown. We have included this new experiment and its quantitative results in the revised manuscript and clarified that the framework is robust to domain mismatch.
>
> Q3. Authors did not carry out any analysis on adaptive adversaries. which is an important concern for practical usage of the proposed approach. Considering adaptive attackers. they may mask stylistic token patterns. this calls out for an ablation on adversarial output perturbation.
>
> A3. We thank the reviewer for pointing out the importance of adaptive adversaries. In the revised manuscript, we added a dedicated Adaptive Adversary Analysis section. We design three adversarial ablations that reflect realistic attacker capabilities: (1) Back-translation rewriting of the training corpus before distillation. (2) Cross-teacher fine-tuning on the same-domain data generated by a different teacher. (3) Inference-time output rewriting to mask stylistic patterns.
>
> As shown in Table 5, these adversarial interventions degrade performance to varying degrees; yet, the auditor remains consistently effective. Back-translation disrupts original stylistic patterns and prevents the model from emitting teacher-specific tokens, thereby substantially weakening token-level stylistic signals. Cross-teacher fine-tuning results in moderate degradation, as conflicting stylistic patterns partially overwrite the original teacher's signature; accuracy drops by approximately 5\%-25\% when LLaMA3 and GPT-4o serve as teacher models. Inference-time rewriting also induces a moderate reduction, with accuracy decreases ranging from 0\% to 25\%, suggesting that even a small subset of preserved model-style tokens is sufficient for correct lineage auditing.

---

> > ### Author Response · Authors · 2025-11-26
> >
> > Q4. How stable are these stylistic tokens across: temperature changes, decoding strategies (greedy, nucleus, beam), etc.?
> >
> > A4. To address the reviewer's concern, we systematically examined the robustness of stylistic tokens under different generation settings, including temperature variations (0.1-1.0) and decoding strategies (greedy, top-k, nucleus sampling, and beam search). This evaluation is conducted only under the word-level access ($K_w$) setting, which estimates token probabilities via repeated generation. The token-level access ($K_t$) setting cannot be evaluated in this manner, as it directly retrieves token probabilities from model logits and does not rely on sampled outputs. Figure 6 shows that extremely low temperatures (0.1 and 0.3) indeed suppress model-specific stylistic signals because sampling becomes nearly deterministic. When using moderate or high temperatures (0.7 and 1.0), which preserve each model's natural generative preferences, the stylistic differences re-emerge, and our method achieves significantly higher accuracy. This demonstrates that our auditor is capturing genuine model-dependent stylistic tendencies rather than artifacts of the decoding procedure.
> >
> > We further evaluated stylistic-token stability under different decoding strategies: greedy, beam search, top-k, and nucleus (top-p) sampling (Figure 7}). Greedy and beam search collapse the distribution toward high-probability modes, suppressing stylistic variability and resulting in low auditor accuracy. In contrast, top-k and nucleus sampling preserve model-specific generative preferences while avoiding deterministic collapse, yielding high accuracy. These findings suggest that stylistic signals are most faithfully retained under commonly used sampling-based decoders, which aligns with modern LLM deployment practices.
> >
> > We have included this ablation experiment in the revised manuscript.
> >
> > Q5. Some unrelated models may coincidentally share stylistic preferences. When does DLI misfire?
> >
> > A5. We appreciate the reviewer's observation that unrelated models may occasionally share stylistic preferences, which could, in principle, cause DLI to misfire. We agree that the most likely scenario for such accidental similarity arises when the suspicious model comes from the same model family or shares highly similar architectural or tokenizer characteristics.
> >
> > To directly test this, we included suspicious models as close as possible to the teacher model, GPT-4o-mini; for example, GPT-Neo-1.3B and GPT-2-Large. Among all unrelated model candidates, they represent the closest case in which stylistic coincidence is most likely to occur. However, DLI still correctly classified these models as non-students. This empirical result strongly supports that (1) Superficial stylistic similarities are insufficient to fool DLI. (2) DLI captures deeper distributional alignment produced by actual distillation rather than mere architectural resemblance.
> >
> > We also conduct some adversarial ablation experiments. As shown in Table 5, when teacher-generated data are rewritten (back-translation) by other models, the stylistic tokens that encode model-specific preferences are often attenuated or erased. Such rewriting operations, paraphrasing, back-translation, or normalization by another LLM tend to smooth out distributional quirks and converge the output toward a more generic style. As a result, the distinct stylistic signatures that DLI relies on become substantially weaker, making it difficult for the auditor to distinguish whether a sample originates from the true teacher model or from an unrelated model.

---

### Meta-Review · Area_Chair_7v57 · 2025-12-18

**Summary:**

While the reviewers agree the paper solves a timely problem in realistic auditing scenarios with strong empirical results, there are also concerns on the method description, overhead/complexity, technical details, analysis, and experiments. Overall the reviewers provided lots of constructive feedback, which the authors thoroughly addressed with many new experiments. However, the amount of new experiments is rather large and calls for the revised paper to be carefully reviewed again. In addition, many experiments only demonstrate that DLI works well in practice instead of really addressing the reviewer's fundamental concerns. For example, Reviewer YXqc pointed out that DLI performance may drop if the prompts are distribution mismatched, and the authors responded that DLI is highly robust even if the prompts are sampled from five distinct domains. This response is not convincing because DLI is not supposed to work properly if the distribution is really different. Other author responses are of this pattern, and perhaps DLI is indeed robust to any problem, but the reviewers have not responded further, which makes the AC think that they were not fully convinced. This is a tough decision, but since the review scores are overall on the negative side, and there is no indication the scores were increased, the recommendation is to resubmit the revised paper.

**Reviewer Concerns:**

The authors provide many new experiments that address most of the reviewer concerns, but several results only emphasize how DLI works well in practice without delving into the fundamental concerns sufficiently. Also experiments on true positive rate (TPR) at low false positive rate (FPR) thresholds are still missing without a clear justification. While the AC respects the significant amount of work by the authors, the problem seems to be that too many things were added to the revision.

**Reviewer Scores:**

There is no indication that the reviewers would have changed their scores.

---

### Decision · Program_Chairs · 2026-01-26

Reject